# An Efficient System Based on Experimental Laboratory in 3D Virtual Environment for Students with Learning Disabilities

**Abir Osman Elfakki** **, Souhir Sghaier * and Abdullah Alhumaidi Alotaibi**

Department of Science and Technology, College of Ranyah, Taif University, P.O. Box 11099, Taif 21944, Saudi Arabia
* Correspondence: sdsghaier@tu.edu.sa

**Abstract:** Virtual reality applications can enhance the education and training of students with learning disabilities, along with their quality of life. Virtual worlds offer opportunities for creating a highly interactive, complex simulation. Modeling and scripting tools can be applied in these worlds. Providing functionalities for managing presentations, administration as well as assessment of coursework in an educational context, virtual learning environments support teaching and learning in schools. Learning disabilities such as "dyslexia, dyscalculia, attention issues (ADHD), and disability in retrieving information", which students need to do physics experiments, are a major source of concern since they impair a person's ability to learn. Traditional teaching methods do not appear to support the same thought in the situation of students with learning disabilities, who normally need a significant investment of time, money, and people for individual tutoring. Students with this problem have difficulty envisioning or visualizing what they are being taught, in addition to breaking down and processing the material. To address the above issues, this paper develops and evaluates a 3D virtual physics laboratory to improve cognitive skills in physics experiments for students with learning disabilities. The environment is created based on specially designed criteria for disabled students. The study proves the effectiveness of the 3D virtual environment in improving the cognitive skills in the physics of students with learning disabilities.

**Keywords:** virtual experiments laboratory; 3D virtual environment; students with learning disabilities; cognitive skills



## 1. Introduction

Virtual Learning Environments (VLEs) are learning tools that incorporate computers and the Internet into instruction to enhance a student's learning experience. Furthermore, they are increasingly being used in a variety of settings, including classrooms, informal learning environments, distance learning, business, and many others. In a virtual world based on a computer-simulated 3D environment, users interact with avatars. This habitat is typically represented in two or three-dimensional forms in graphical representations of humanoids [1]. Virtual Environments (VEs), in addition to being interactive, engaging, and safe, typically provide students with immediate feedback, allowing them to learn by doing. Because of their importance and potential, 3D virtual platforms created with the OpenSimulation tool rank high among educational VEs.

The ability to integrate knowledge and skills with educational VLEs and exchange information with intellectual information management will be critical in increasing the adoption of Virtual Reality (VR). On the other hand, VR can assist students with disabilities in developing their knowledge, abilities, and behaviors in different manners, which would not have been possible otherwise, allowing them to engage in academic tasks that are largely free of the restrictions set by their disability and totally secure. VR also tends to help everyone else feel compassion for people with disabilities by enabling them to practice disabilities through virtual worlds [2]. According to the American Psychiatric Association, the classification of students with disabilities is ten categories: intellectual disabilities, Autism

Spectrum Disorder (ASD), attention-deficit/hyperactivity disorder, global developmental delay, communication disorders, specific learning disorders, developmental coordination disorders, stereotypical movement disorders, tic disorders, and Tourette's syndrome [3,4]. Among these disorders, the highest rates are seen for intellectual disability and ASD [5]. Besides offering a compelling and exciting experience, these applications help students with learning disabilities reduce the impact of their disabilities, enhance their quality of life, get involved in societies, learn how to live, be more accessible, and develop their cognitive abilities [6]. The design of VEs can be adapted to meet the needs of students with a wide range of educational backgrounds, physical capabilities, verbal capabilities, and cognitive abilities [2].

In contrast, students with learning disabilities can discover or build new worlds or control items without being restricted by their disabilities if they are provided with an appropriate interface that gives students a sense of control in their surroundings through VR freedom of mobility [6]. Furthermore, a VE is more personal than most real-life environments because it can be tailored to the abilities of the individual, allowing individuals with disabilities to participate [6]. It is possible to use VR to engage students with learning disabilities, focusing on their talents and enabling them to gain a sense of mastery [7]. In our paper, we are therefore attempting to prove that VR can benefit students with learning disabilities.

## 2. Literature Review

### 2.1. The Benefits of VR for Students with Disabilities

A virtual reality environment simulates a three-dimensional (3D) computer-generated environment, which lets the user explore, interact, and experience it physically, usually via special electronic equipment. Head-Mounted Displays (HMDs) or glasses are worn by individuals. In addition to the audio and video input, as well as the virtual experience, many HMDs come with speakers or headphones. Input devices such as joysticks and trackballs are also widely available. Individuals can use VR as an assistive technology by lessening or offsetting the effects of their disability and by providing an alternative method of accomplishing their goals. Through the creation of new environments, children with disabilities can manipulate objects and experience things that would normally be impossible or difficult in real life for them. VR for rehabilitation, such as cognitive evaluation, has also been demonstrated to be beneficial [8]. Students with learning disabilities can also benefit from VR through its social benefits. Students can feel empowered and interact with other students with similar problems or conditions through VR and gain a feeling of self and power. Instead, students will concentrate on their self-sense rather than their impairment by using avatars (a virtual character that a student wishes to assume). VR may provide students with a new perspective, the option of exploring diverse viewpoints, or the opportunity to assume different identities. Talking with other online users within a VE equalizes the student and gives a social activity for students who would otherwise have been alone.

### 2.2. Personalizing VR to Fit the Needs of Students with Disabilities

VR has characteristics that make it an ideal platform for creating adaptive education programs for disabled students, especially due to its adaptability. It is possible to customize the VE properties so that certain stimulus types can be included or excluded, depending on the service purpose. This adaptation aids students with disabilities in having the best possible interaction. By transferring information from the damaged senses into information that the intact senses can see, VR allows students with sensory impairments to experience what would have been difficult or impossible for them otherwise. For students who are blind, a VE using auditory and/or tactile cues can be implemented. Students with autism benefit from VEs which contain a limited number of stimuli to help them focus. As the student's time to complete a task increases, the number of simultaneous stimuli can be gradually raised until the students can function at a level that is comparable to real-world

situations [9]. It is possible to deconstruct complex activities into smaller, more manageable parts for students with learning disabilities until they master the required skills in the same way. Almost no need for concepts, symbols, or linguistics is required. As a result, VR is more accessible to a wider range of consumers who could benefit from learning a job in VR that is free of the constraints of conventional teaching methods. Besides, VR has a significant effect on the learner's special needs, as mentioned in [10,11].

### 2.3. Challenges of Virtual Learning Environments for Students with Learning Disabilities

In addition to the lack of staff and resources, current VEs cannot cope with three critical problems (other than those related to the teaching-learning process).

- Several Special Education centers still have limited access to the Internet today. In these situations, ignoring the fact that practically all institutions now have appropriate Internet access, certain countries still have issues with connectivity, bandwidth, or even a complete Internet connection.
- Lack of instruments to study the impact and acceptance of virtual/digital educational materials that are not appropriately designed to work with children with various types of disabilities such as cerebral palsy, intellectual disability, Down Syndrome, and so on [11].
- In terms of content and virtual learning settings, Special Education professionals and children with disabilities.

### 2.4. Criteria for Designing VR Environments for Students with Learning Disabilities

There is a set of criteria derived from the results of research, studies, and theories in the field of education and psychology which should be considered when designing and producing a VR environment for children with learning disabilities.

- ➢ A study by [12] that aimed to determine the criteria for designing VR environments for people with learning disabilities also found the following:
  - (a) Determining the needs of the children;
  - (b) Usability;
  - (c) Analyzing the tasks, breaking them down into subtasks, and assigning them out.
- ➢ A study by [13] aimed at determining the requirements necessary for designing a VR environment for people with learning disabilities, with the following:
  - (a) The preferred desktop VR system is for students with disabilities;
  - (b) A child with learning disabilities would benefit from using a mouse, which is one of the best interaction devices in VEs.
- ➢ According to a study by [14], VR can be used by students with learning disabilities as follows:
  - (a) The user interface should be comfortable with technical assistance.
  - (b) Entry tools should be comfortable and conform to health and safety standards to avoid child stress.
  - (c) A child will be less likely to be violent towards written information if attractive colors attract their attention.
- ➢ According to the study of [15,16], a usable design should:
  - (a) Be simple to use and comprehend;
  - (b) Follow conventions or standards;
  - (c) Be flexible and effective in prompting and providing feedback.

### 3. Related Work

There have been several studies on this topic since the late 1990s. Among them, we mention

- Current uses of VR for children with disabilities [17]. In this paper, a desktop VR (VR) program was designed and evaluated to teach children about the accessibility and attitudinal barriers encountered by their peers with mobility impairments. For example, within this software, children sitting in a virtual wheelchair would experience obstacles such as stairs, narrow doors, objects too high to reach, and attitudinal barriers such as inappropriate comments.
- The effect of 3D VR on sequential time perception among deaf and hard-of-hearing children [18]. This study examined whether deaf and hard-of-hearing children perceive a temporal sequence differently under various representational modes. They compared the effect of three-dimensional (3D) VR (VR) representations on sequential time perception among deaf and hard-of-hearing children with pictorial, textual, spoken, and signed representations.
- 3D sound interactive environments for blind children's problem-solving skills [19]. This study presented the design and usability of 3D interactive environments for children with visual disabilities to help them solve problems in Chilean geography and culture and introduced Audio Chile. This virtual environment could be navigated through 3D sounds to enhance spatiality and immersion throughout the environment.
- A review on VR for educating students with learning disabilities [20]. This paper described various types of university laboratories from around the world to present the most recent state-of-the-art. A discussion of the educational effectiveness of each laboratory format was also included. Finally, a case study of Technological de Monterrey's development and implementation of VR laboratories and remote labs was presented.
- Virtual Lab Model for Making Online Courses More Inclusive for Students with Special Educational Needs [7]: The creation of a laboratory model that might use available online courses and adapt them for students with unique educational needs was prioritized. The study looked at how students would learn in a virtual lab, how to integrate online courses, and how to deliver them to students with specific educational requirements.
- Sghaier et al. in [21] develop an expert system for meta-learning. This intelligent system is self-configurable according to the different users of the metaverse. Their system based mainly on the real-time learning, and training of disables students in virtual worlds and 3D space especially in the current COVID-19 pandemic situation. It aims to design and create a virtual learning environment, using (Open Simulator) which is based on a 3D virtual environment using simulation of the real word environment. Then they connect it to the learning management system (MOODLE) through technology for 3D virtual environments (SLOODLE). That is, providing the ability to manage disabled student records and follow up on their activities, tests, and exams and has the advantage of the ability to learn systems to store educational content.

## 4. Methodology of the Proposed Approach

The purpose of this study is to establish an interactive communication platform for teachers and students with learning disabilities in the context of physics experiments using a 3D virtual physics laboratory. The authors propose to develop and evaluate a 3D VR laboratory to improve cognitive skills in physics experiments for students with learning disabilities. A system must provide us with the necessary technology and tools to incorporate an emulator of the 3D virtual laboratory (OpenSim) that is used to simulate it. By connecting the virtual environment with the learning management system, Moodle would allow students to manage the latter from inside the virtual physics lab. In other words, open-source software was used, including OpenSimulator as well as LMS (MOODLE), which was integrated with OpenSim through SLOODLE technology. Furthermore, the SLOODLE project was an open-source project aimed to integrate web-based learning features, allowing a VR laboratory to the 3D virtual environment available on the web. The system will be available as an application on a large internal network or over the web [22].



### 4.1. Functional Requirements Provided by OpenSimulator or SLOODLE

- The educational content of the learning management system (Moodle) is easily created and managed inside the 3D virtual environment and the virtual physics laboratory. It interacts with different users in the 3D virtual environment and virtual physics laboratory.
- The user of the 3D virtual physical laboratory is not allowed to modify the content of the 3D virtual environment but just interacts with it.
- The access authorization and available permissions must allow access to the 3D virtual physical laboratory according to specific authorities, besides the initial access to the 3D virtual environment via a username and password.
  - (a) The 3D virtual physical laboratory enables and provides several functions and capabilities to the user and virtual personality, including inviting friends to visit their virtual learning physical laboratory and 3D virtual environment, controlling the virtual personality and its appearance, walking, flying, teleporting, sending group messages, and listening to group conversations, etc.
  - (b) The 3D virtual physical laboratory offers and provides us with a dedicated space for audio conferencing and sound support and integrates them into the 3D virtual environment, in addition to the automatically integrated sounds compatible with this type of educational 3D virtual environment to enable complete immersion in the experiment, where 3D audio effects are necessary.

### 4.2. Development Requirements or Creative Solutions of Our Proposed Approach

- Our proposed approach manually configures the learning management system to integrate with SLOODLE tools by copying and pasting SLOODLE configured files into the learning management system to effectively implement them together. Then, we create and design the 3D primitive shapes used for experiments. Finally, based on the characteristics of the experiment, we program each object with an LSL script. Moreover, our proposed approach to developing a 3D virtual learning environment contains the following functions:
- The 3D virtual physical laboratory links the name of the user or virtual character in the (OpenSim) emulator to the same user in the learning management system (Moodle) through the email service, as the virtual character enters their account in the learning management system (Moodle) from the 3D virtual environment.
- It interacts with the various tools of SLOODLE in the 3D virtual environment.
- A certain attribute of a 3D virtual physics laboratory considers a particular criterion for enabling students with disabilities to learn with special content. The criteria include the color for the primitive 3D model shapes (balls, cones, cubes, etc.), the way 3D models fall in accordance with physics laws, students' attention to interactive sounds in the 3D virtual learning environment, and the flying features which enable the user to fly around the 3D virtual laboratory especially when it comes to an understanding of the rotation and motion of the planet for students with learning disabilities. These criteria are summarized as follows:
  - i. Using attractive colors will catch the child's attention and reduce their tendency to be violent with written information.
  - ii. Notecards are used to demonstrate the steps of the experiment and explain them.
  - iii. Students with learning disabilities can display presenter tools, which allows the student to show whole physics experiments as PDF files, web pages, and videos.
  - iv. Students with learning disabilities can use a glossary tool that Moodle students can use to identify terms. To obtain immediate results, students or teachers must enter the word "/def" followed by the name of the term for which they want to search, which is registered in the glossary.

Table 1 presents the different criteria for the design of the proposed experimental laboratory with diverse categories.

**Table 1.** Design criteria in 3D virtual physics laboratory.

| Category | Design Criteria Available in Our Virtual Physics Laboratory |
|---|---|
| VLE facilities | Simple login system for VLEs.<br>A simple way to launch programs. |
| Tutor information and control | Indications of the appropriate educational level and tutor info.<br>Set up is simple for each student. |
| 3D virtual learning environment content | Identifying learning materials (experiments) with students.<br>Making learning fun.<br>Making learning interesting.<br>Multiple versions to suit different abilities.<br>Enabling students to be as effective as mainstream users. |
| Interaction facilities | Providing speech output.<br>Exploring elements in the 3D virtual physical laboratory.<br>Designing the virtual laboratory to fit the capabilities of students.<br>Easy to interact with objects in the virtual environment. |

*4.3. Proposed Virtual Physical Laboratory Architecture*

We need a flexible design that can meet all the specifications, as shown in Figure 1. The 3D virtual physical laboratory architecture style is made up of three parts; the first part is the simulation tool (OpenSimulator), which is the virtual educational 3D virtual environment world. The second part is the SLOODLE software, which is the link that links the 3D virtual environment and the learning management system. Lastly, there is the learning management software, the delivery method for e-learning.

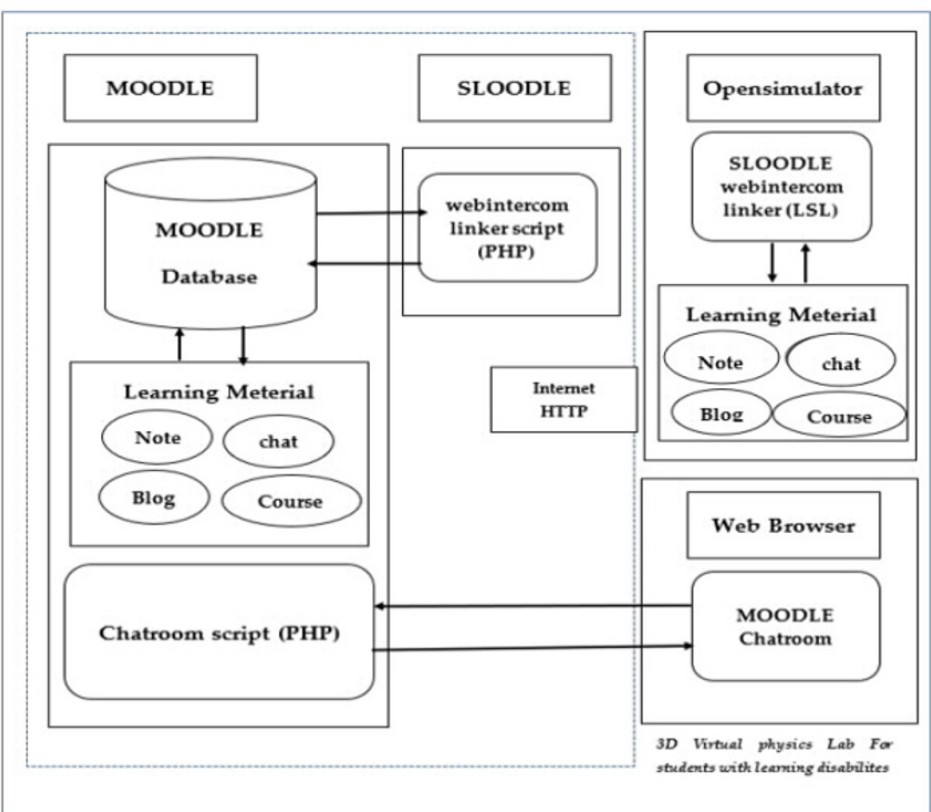

**Figure 1.** Proposed virtual physics laboratory architecture.

*4.4. Techniques and Tools Used in the Implementation of Virtual Physical Laboratory*

Five tools/techniques are used to implement and develop this intelligent system (Figure 2), which is explained below.

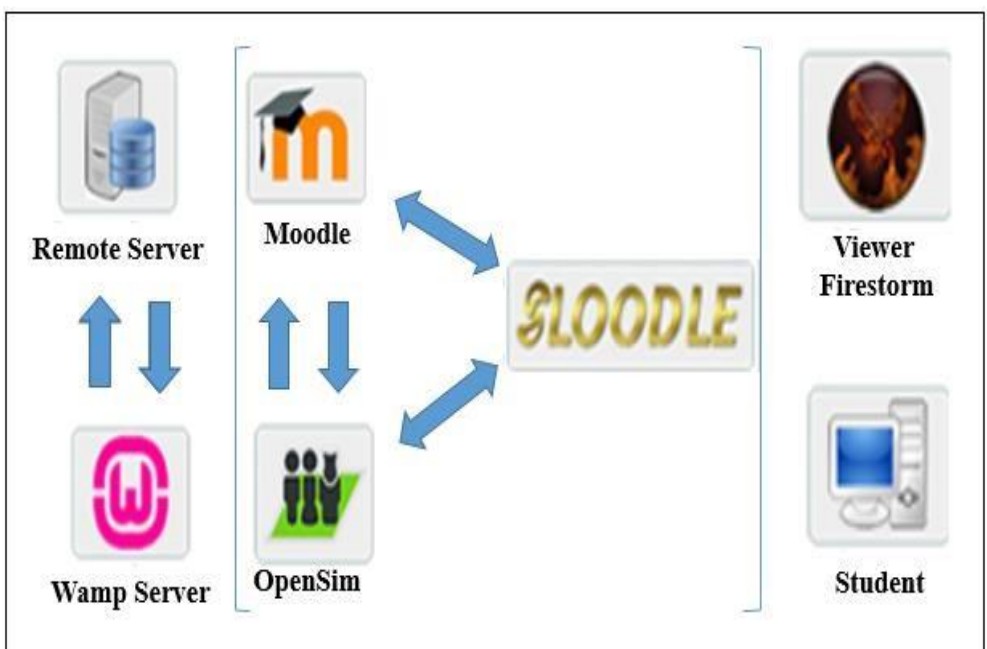

**Figure 2.** Tools and techniques used to implement the system.

- Wamp Server: This server is chosen because it is open source for the Windows operating system and hosts the applications required to run the technologies. These include databases (MYSQL), a programming language (PHP), and a server (Apache), and it creates a local server on your machine that hosts the database application (MYSQL), the 3D virtual environment OpenSimulator, and the learning management system databases (Moodle).
- The Moodle learning management system was selected because it is an open-source application widely used in educational institutions and institutes. In addition, the learning management system (Moodle) materials and activities are well connected to a network available to students and teachers through the SLOODLE technology.
- SLOODLE technology/learning management system: It is chosen because it is an open-source technology that allows integration between the 3D virtual environment OpenSimulator and the learning management system (Moodle). This allows you to view and interact with the materials and activities of the learning management system (Moodle).
- Firestorm Viewer: It is selected because it is an open-source application, and it has features that allow communication with the world and the 3D virtual environment, as well as the possibility of importing several 3D objects.
- OpenSimulator server: It is chosen because it is an open-source application. It is a server that allows us to create, design, and emulate a 3D virtual environment to which the 3D virtual environment browser is connected in order to display the 3D virtual environment. OpenSimulator can be used to create virtual worlds like Second Life. OpenSimulator, on the other hand, does not intend to be a clone of the Second Life server platform. Rather, the project goal is to enable the development of innovative features for VEs and the metaverse as a whole. The history of OpenSimulator is intertwined with the development and deployment of the Second Life platform [23]. Linden Labs, Inc [24] owns and distributes Second Life, which consists of two key components: a server that hosts a virtual world comprised of a massive rectangular grid of regions and a client program that runs on the user's computer and communicates with the server [25]. The functions of the client and server are distinct [26].

## 5. Implementation

Our proposed study provides an experimental 3D virtual physics lab for students with learning disabilities, as shown in Figure 1, with enough flexibility that allows them to learn without stress. The students accepted the virtual laboratory and interacted with it very well. They have made progress in using the system collaboratively. We collect and analyze data to verify their subjective ratings in the post-survey. Students with learning disabilities use sound because it simplifies custom coordination and communication. They also use simulation and 3D models. They do not have any problems with the visual complexity of 3D models, as it initiates interaction with them within the virtual environment through various commands. As shown in Figure 3, the 3D virtual lab contains a set of three-dimensional models that could be practiced in addition to conducting various experiments, such as determining the speed at which a fall occurs with a difference in the reaction of falling. The following three-dimensional models are evaluated, depending on their shape and the height at which they fall: a cube, a ball, and a cone, as the virtual lab allows the programming of objects and three-dimensional models according to the laws of physics. The students also enjoy the three-dimensional representations, particularly those showing the solar system, where they learn about the planets, the distances between them, and the speed of their rotation around their axes and the sun because this gives them a lively interactive presentation, where full information about each planet is displayed as soon as the student approaches it. These interactions are programmed inside the lab using a special script called LSL for 3D VEs. The students also use a virtual slide projector to read the details of the scientific experiment. They also use the SLOODLE's Choice tool, which enables the student to vote and express his opinion about the virtual lab of physics and the extent to which it is beneficial. The vote will appear within the Moodle platform. The board is labeled "unnecessary". Text chat is largely overlooked; audio is better and more efficient. Users personally prefer to interact via headphones to keep themselves from their physical environments. In turn, they know each other and work together. They have common ground and a trusting relationship that balances the need for a direct visual link with their peers. Students with learning disabilities are able to finish most of the basic notes in a timely manner.

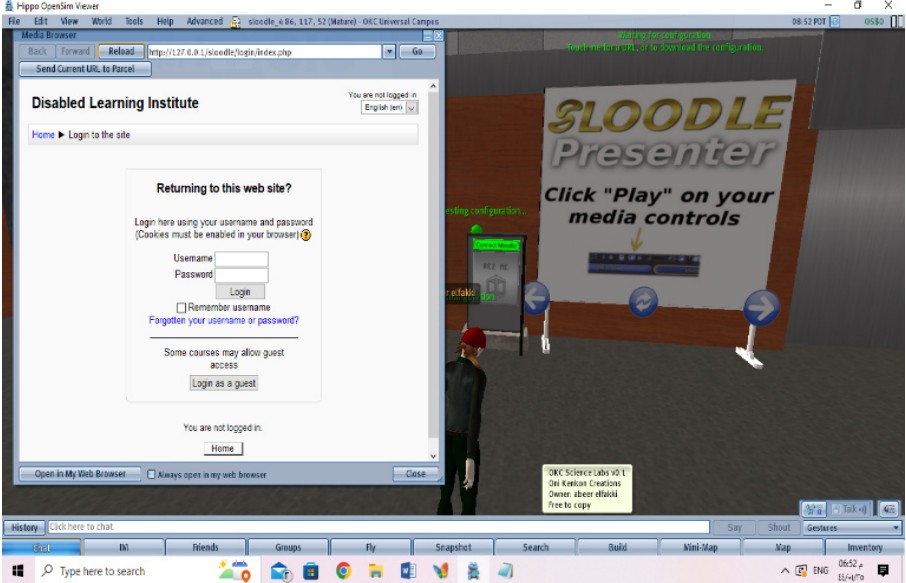

(**a**) Connection to the learning management system (Moodle) via Sloodle

**Figure 3.** *Cont.*

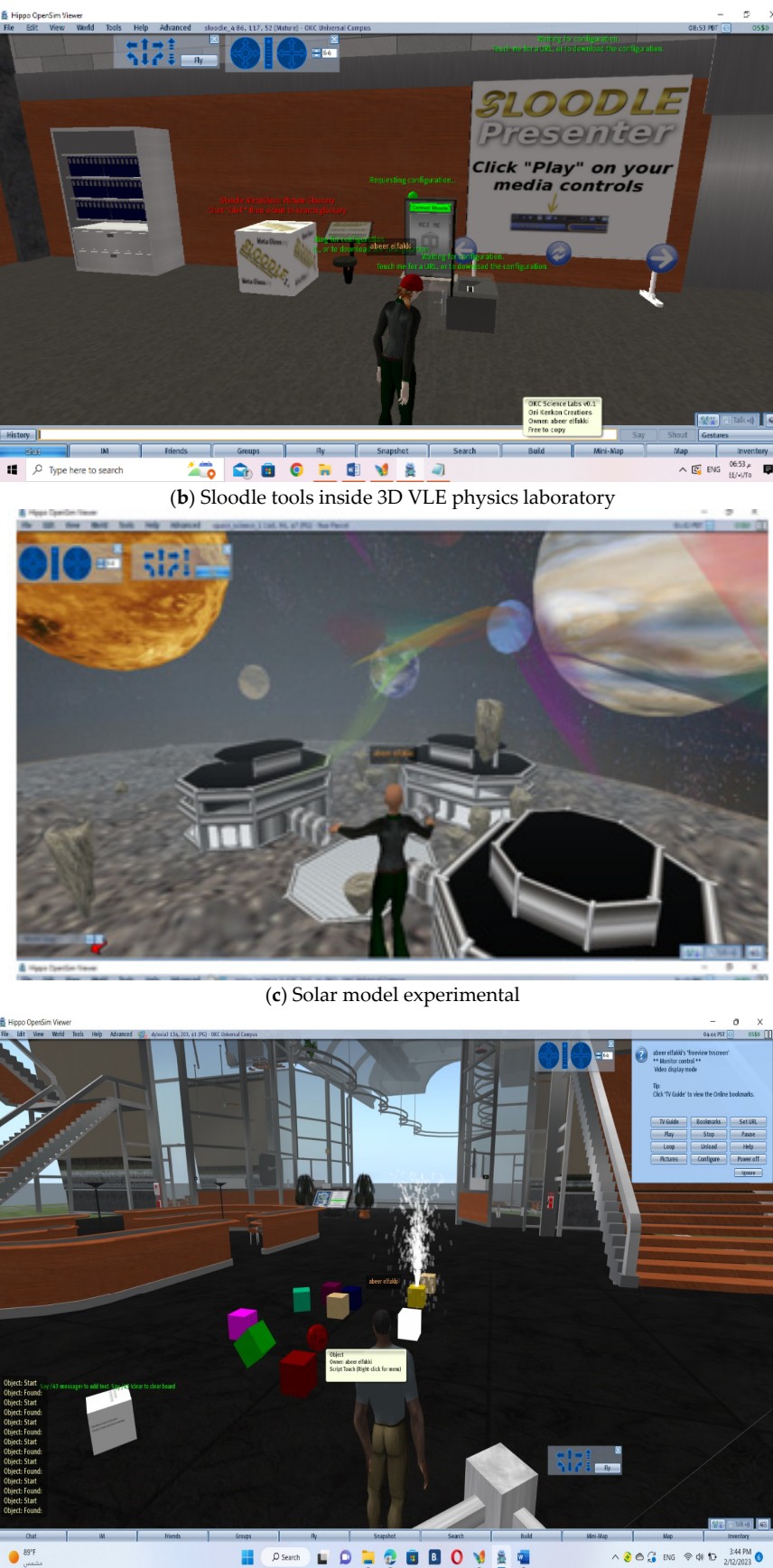

(**b**) Sloodle tools inside 3D VLE physics laboratory

(**c**) Solar model experimental

(**d**) Experimental of Physical simulations in 3DVLE

**Figure 3.** Experimental examples of proposed system.

### 5.1. Operational Phases

Based on the analysis and design proposal of the system, the system is implemented in stages.

(a) The first phase: Customizing the learning management system (Moodle) setups. In the initial step, a virtual class is established in the learning management system (Moodle), and all the learning management system's operations, such as chatting, assignments, exercises, a glossary, and others, are necessarily linked to three-dimensional objects within the built and developed 3D virtual environment. This stage of implementation allows the teacher to transform and transfer their educational content to the virtual three-dimensional 3D environment. For example, the teacher can create a glossary of terms and chat rooms for specific subjects and give students specific exercises and assignments within these rooms.

(b) The second phase: Establishing the virtual lab building. In this phase, a lab building is constructed, besides the allocation of equipment in it, such as a room for presentations. In addition, we put information about the tool used inside the room.

(c) The third phase: Installing the technology configurations SLOODLE. At this stage, the technology configuration files SLOODLE must be a part of the educational management system (Moodle) directories for the supervisor to add technology activities from within the learning management system (Moodle), and the two have become integrated with the 3D virtualized environment which is accessible and enabled.

(d) The fourth phase: Start creating technical resources SLOODLE. Here, the software is identified, and its setup is installed. For instance, we select an enrollment booth to assist with registering virtual characters in the learning management system (Moodle). Furthermore, the Presenter tool facilitates the presentation established in the Moodle chatting tool (Web-Intercom) to connect discussion forums in two 3D VEs, in addition to other tools of SLOODLE.

(e) Fifth phase: Completing the setup of all SLOODLE capabilities. At this phase, all capabilities are connected to the learning management system (Moodle) and fully prepared to be used.

### 5.2. D Virtual Physics Laboratory Experiments

The virtual physics lab deals with a number of experiments related to the fall of objects from various heights in the air, noting the speed of the reactions of the fall depending on the shape and weight of the body according to Newton's laws of motion, where the student may interact with these bodies of different shapes, if they can, through touch. The body can interact with it by touching and increasing its size and weight to test the movement of its fall with the difference in size and weight. These three physical laws establish the science of body movement and link the forces affecting the body's movement. Furthermore, an experiment is related to the movement of the planetary rotation because the orbit of each planet is an ellipse, and the sun is in one of its foci, as described by Kepler's first law of planetary motion. Close movement, in which the student can be present near these planets and interact with them according to the solar system model, allows displaying information about the planets, their rotation speed, and their distance from the sun and the rest of the planets by interacting with them. This information appears as soon as the student approaches the planet. Using the LSL script, special effects are programmed by adding details about the planets to the script. These experiments are carried out over two weeks at the end of the semester, with two lessons per week lasting 40 min. During this time, pre-tests are also carried out in which experiments are done without the use of a physics laboratory. The virtual classroom is created through explanations and watching videos about these experiments. Then, a pre-test for the students is applied to these experiments. After that, the experiments are conducted using the virtual physics laboratory. Finally, a post-test on these experiments is performed. The results show an improvement in the student's performance in post-tests compared to pre-tests. As shown below, these forms contain an explanation and details of Newton and Kepler's laws:

(a)  Kepler's laws of planetary motion

First, all planets orbit the sun in the form of an ellipse. Mathematically, an ellipse can be represented by the following formula:

$$r = \frac{\rho}{1 + \varepsilon \cos \theta} \tag{1}$$

(b)  Newton's first law of motion of a body

Unless compelled to change its state by forces impressed upon it, everybody remains at rest or in uniform motion in a straight line.

$$\sum F = 0 \iff \frac{dv}{dt} = 0 \tag{2}$$

(c)  Newton's second law of body motion (Circular motion)

Second, an object's motion is proportional to the force impressed and occurs in the direction of the straight line in which the force is impressed.

$$\vec{F} = \frac{\vec{d_p}}{d_t} \tag{3}$$

(d)  Newton's third law of body motion

There is always an equal reaction to every action, or the mutual actions of two bodies on each other are always equal and directed to opposite parts.

$$\frac{\vec{d_p}}{d_t} = \frac{\vec{d_{p1}}}{d_t} + \frac{\vec{d_{p2}}}{d_t} \tag{4}$$

## 6. Results and Findings

### 6.1. Consistency Analysis of Different Groups before Experiment

To properly assess our proposed 3D virtual physics laboratory, we use a study sample of 40 female students with learning disabilities around many middle schools in RANYAH—TAIF, which are randomly selected. The study sample is split into two groups: the experimental group, which uses the three-dimensional virtual environment and includes 20 students with similar learning abilities with respect to physics before the experiment, and the control group, which uses the conventional process, includes 20 students. Students with learning disabilities are divided into two equal groups (n = 20 each) based on their disability type. Students with attention deficits, for example, are given experiments on three-dimensional objects with appealing colors. Students with poor concentration and difficulties with mathematical operations are given planetary rotation motion experiments in the solar system in addition to motion experiments. For various bodies, reactions are based on the body's initial starting speed until it stops, with the provision of functions that calculate the body's speed based on its mass. The conventional process consists of various experimental displays, such as videos about planets' motion and rotation, as well as some 3D models, like cubes, balls, and cones, used to execute practical experimental physics.

Cognitive assessment of practical experience skills. Based on Bloom's four levels of knowledge, this test aims to measure the cognitive aspect of practical experience skills for middle school students in the city of TAIF-RANYAH. According to Bloom's four levels of knowledge: remember, understand, apply, and analyze, we create a table of specifications for the test, which originally consisted of 16 items. The specifications of the cognitive test for practical experience skills are shown in Table 2.

**Table 2.** Specifications of cognitive test for practical experience skills.

| Subject | Behavioral Goal Levels | | | | Sum |
|---|---|---|---|---|---|
| | Analyze | Apply | Understand | Remember | |
| Newton's first law of body motion | 0.5 | - | - | 0.5 | 1 |
| Newton's 2nd law of body motion (Circular motion) | 1 | 0.5 | 0.5 | 1 | 3 |
| Newton's third law of body motion | 0.5 | 1 | - | 0.5 | 2 |
| Kepler's laws of planetary motion | 1 | 1 | 2 | - | 4 |
| Summation | 3 | 2.5 | 2.5 | 2 | 100% |

Table 2 summarizes an assessment of the proposed system.

6.1.1. Study Assignments

This study aims to verify the validity of the following five hypotheses:

(a)  First hypothesis: there is a statistically significant difference at the level of significance ($\alpha = 0.05$) between the mean scores of the experimental group that uses a 3D virtual physics lab and the mean scores of the control group that uses the traditional method in the post-measurement of the cognitive test of practical experience skills in favor of the experimental group.

(b)  Second hypothesis: there is a statistically significant difference at the level of significance ($\alpha = 0.05$) between the mean scores of the experimental group that uses a 3D virtual physics lab and the mean scores of the control group that uses the traditional method in the post-measurement of the performance test for practical experiment skills in favor of the experimental group.

(c)  Third hypothesis: there is a statistically significant difference at the level of significance ($\alpha = 0.05$) between the mean scores of the experimental group that uses a 3D virtual physics lab and the mean scores of the control group that uses the traditional method in the post-measurement of practical experiment skills in favor of the experimental group.

(d)  Fourth hypothesis: there is no statistically significant difference at the level of significance ($\alpha = 0.05$) between the mean scores of the experimental group that uses a 3D virtual physics lab and the specific degree of mastery, 80%, in the skill test of practical experiment skills.

(e)  Fifth hypothesis: there is no statistically significant difference at the level of significance ($\alpha = 0.05$) between the mean scores of the control group that uses the traditional method and the specific degree of proficiency, 80%, in the skill test of practical experience skills.

To determine the significance of the differences between the averages of the experimental and control groups in the post-measurement cognitive test of practical experience skills, we calculate a t-test for two separate samples, as provided in Table 3.

**Table 3.** Significance of differences between mean scores of experimental group and control group in post-application of cognitive test of practical experience.

| Groups | SMA | Standard Deviation | Sample (n) | Value (T) | Value (Sig) | Indication Level |
|---|---|---|---|---|---|---|
| **Control group** | 11.15 | 0.688 | 20 | 13.838 | 0 | Function |
| **Experimental group** | 15.50 | 1.226 | 20 | | | |

Table 3 shows the high level of achievement of the experimental group of students who used the 3D virtual physics lab compared to the control group who used the conventional process. The experimental group's average ratings are 15.50, while the control group's

average ratings are 11.50. The "t" significance is 13.838, and the significance level is 0, which is recognized as a significant statistical value. In fact, the experimental group has a higher average. As a result, the first hypothesis is adopted. We also perform a t-test on two separate samples to assess the significance of the variations between the experimental and control group averages in the post-measurement of the performance test for practical knowledge, as depicted in Table 4.

**Table 4.** Significance of differences between experimental and control group averages in post-application of performance test for skills of practical experiments.

| Groups | SMA | Standard Deviation | Sample (n) | Value (T) | Value (Sig.) | Indication Level |
|---|---|---|---|---|---|---|
| **Control group** | 101.30 | 2.203 | 20 | | | |
| **Experimental group** | 74.40 | 6.012 | 20 | 18.788 | 0 | Function |

Table 4 compares the high level of achievement of the experimental group that applies the three-dimensional virtual environment to the control group which applies the conventional process. The value of "t" is 18.788, and the value of sig is 0, both of which are statistically significant. Indeed, the higher group is the experimental group.

We also calculate a t-test for two separate samples to determine the significance of differences in the average values of the experimental and control groups in the post-measurement of practical experiment skills, as given in Table 5.

**Table 5.** Significance of differences between averages of experimental and control group in post-application of skills of practical experiments.

| Groups | SMA | Standard Deviation | Sample (n) | Value (T) | Value (Sig) | Indication Level |
|---|---|---|---|---|---|---|
| **Control group** | 116.80 | 2.331 | 20 | | | |
| **Experimental group** | 85.55 | 5.781 | 20 | 22.422 | 0 | Function |

Table 5 demonstrates that the experimental group of students who participated in the 3D virtual physics lab outperformed the control group who used the conventional process. The value of "t" is 22.422, and the value of sig is 0, which is recognized as a statistically significant value in favor of the experimental group with the highest mean. Therefore, the third hypothesis is confirmed.

The significance of the differences between the mean scores of the experimental groups and the specific degree of mastery of 80% in the skill test for practical experience skills is determined using a t-test of linked samples, as shown in Table 6.

**Table 6.** Significance of differences between average scores of experimental group and specific degree of proficiency 80% in skill test of practical experiment skills.

| Groups | SMA | Standard Deviation | Sample (n) | Value (T) | Value (Sig) | Indication Level |
|---|---|---|---|---|---|---|
| **Experimental group** | 116.80 | 2.331 | 20 | | | |
| **Proficiency score 80%** | 98.00 | 0.000 | 20 | 36.075 | 0 | Function |

Table 6 shows that the skill level of the experimental group students increased more than the specified degree of proficiency of 80%, as the average scores of the experimental

group in the dimensional measurement are 98%, which is greater than (80%). The value of "T" is 36.075. The sig value is 0, a statistically significant value at the $\alpha = 0.05$. Thus, the fourth hypothesis is accepted.

We use the t-test of the linked samples to determine the significance of the differences between the mean scores of the control group and the specific degree of proficiency (80%) in the skill test for practical experience skills, as presented in Table 7.

**Table 7.** Significance of differences between average scores of control groups and specific degree of proficiency 80% in skill test of practical experiment skills.

| Groups | SMA | Standard Deviation | Sample (n) | Value (T) | Value (Sig) | Indication Level |
|---|---|---|---|---|---|---|
| Telemetry | 85.55 | 5.781 | 20 | | | |
| Proficiency score 80% | 70.00 | 0.000 | 20 | 9.631 | 0 | Function |

The results of Table 7 show that the skill level of the students of the control group decreases, as it does not reach the required level. The average score of the control group in the post-measurement is 70%, which is less than 80%, and the value of "T" is 9.631. The sig value is 0, a statistically significant value at the $\alpha = 0.05$ level. Thus, the fifth hypothesis is rejected.

6.1.2. Measuring the Effectiveness of 3D Virtual Physics Laboratory in Developing Practical Skills

The current study seeks to achieve the following objectives:

➢ First question: measuring the effectiveness of the three-dimensional virtual environment SLOODLE in developing the cognitive aspect of practical experiment skills in physics for middle school students in the second grade in Taif—Ranyah.
➢ Second question: measuring the effectiveness of the three-dimensional virtual environment SLOODLE in developing the performance aspect of the skills of practical experiments in physics for middle school students in the second grade in Taif—Ranyah.
➢ Third question: measuring the effectiveness of the three-dimensional virtual environment SLOODLE in developing the skill side of practical experiment skills in physics for middle school students in the second grade in the city of Taif—Ranyah.

To measure the effectiveness of the physics lab inside the 3D virtual physics laboratory in developing the cognitive aspect of practical experiment skills in physics for intermediate students who suffer from learning disabilities in Taif. To achieve this purpose, we use Black's adjusted gain Equation (5). The adjusted earnings ratio is

$$MG = \frac{M_2 - M_1}{P - M_1} + \frac{M_2 - M_1}{P} \tag{5}$$

where $M_2$ is the dimensional mean, $M_1$ is the tribal mean, and $P$ is the maximum score for the test.

The adjusted gain ratio is used in Table 8 to measure the efficiency of a physics lab inside the 3D virtual physics laboratory in developing the cognitive aspect of practical experiment skills in physics for first-grade intermediate students with learning disabilities in Taif—Ranyah.

**Table 8.** Adjusted gain ratio.

| Application | Average | Finale Grade | Adjusted Gain Ratio |
|---|---|---|---|
| Pre-application | 2.75 | | |
| Post-application | 15.50 | 106 | 1.75 |

It is evident from Table 8 that the average gain percentage is 1.759, which is more than the minimum set by Black (1.2). Therefore, the three-dimensional virtual environment is considered effective in developing the cognitive aspect of practical experiment skills in physics for intermediate students with learning disabilities in Taif. Thus, the first question of the study is answered. A 3D virtual physics laboratory in developing the performance aspect of practical experiment skills in physics for intermediate students who suffer from learning disabilities in the city of Taif. To achieve this purpose, the modified "Black" gain equation is used. The following table shows the adjusted gain ratios of Black to measure the effectiveness of the 3D virtual physics laboratory in developing the performance aspect of practical experiment skills in physics for intermediate students who suffer from learning disabilities in the city of Taif—Ranyah in Saudi Arabia.

The black adjusted gain ratios used in Table 9 for measuring the effectiveness of the 3D virtual physics laboratory in developing the performance aspect of practical experiment skills in physics for intermediate students with learning disabilities in Taif city. It is evident from Table 9 that the average gain ratio is 1.68, which is more than the minimum set by Black (1.2). Therefore, the three-dimensional virtual environment is considered effective in developing the performance aspect of practical experiment skills in physics for middle students with learning disabilities in Taif. Thus, the second question of the study is answered.

**Table 9.** Black adjusted gain ratios.

| Application | Average | Finale Grade | Adjusted Gain Ratio |
|---|---|---|---|
| Pre-application | 23.05 | 106 | 1.68 |
| Post-application | 101.30 | | |

Table 10 shows the adjusted gain ratios of Black to measure the effectiveness of the 3D virtual physics laboratory using SLOODLE in developing the skills of practical experiments in physics for intermediate students with learning disabilities in Taif.

**Table 10.** Black's modified gain ratios.

| Application | Average | Finale Grade | Adjusted Gain Ratio |
|---|---|---|---|
| Pre-application | 25.80 | 122 | 1.69 |
| Post-application | 116.80 | | |

Black modified gain ratios in Table 10 measure the effectiveness of the 3D virtual physics laboratory in developing the skills of practical experiments in physics for intermediate students with learning disabilities in Taif city.

It is evident from Table 10 that the average gain ratio is 1.69, which is more than the minimum set by Black (1.2). Thus, the three-dimensional virtual environment is considered effective in developing the skills of practical experiments in physics for intermediate students with learning disabilities in Taif. Therefore, the third question of the study is answered.

## 7. Discussion of Achieved Results

We created our virtual physics laboratory based on the simulation model repository files (OpenSimulator) installed through our simulation software file system. Pre- and post-tests are conducted on the same subjects, as well as physical experiments for both groups of students with learning disabilities. As a result of using the 3D virtual physics laboratory, female students with learning disabilities performed well in the study, showing how effective VR laboratories are for learning-disabled students. This illustrates the effectiveness of the SLOODLE 3D virtual physics laboratory, as depicted in Tables 11 and 12.

**Table 11.** Summarized achieved results of cognitive assessment of 3D virtual physics lab.

| Subject | Aspect | Results | |
| --- | --- | --- | --- |
| | | Experimental Group | Control Group |
| Post-application | Cognitive test | 15.50 | 11.15 |
| | Performance | 101.30 | 74.40 |
| | Practical skills | 116.80 | 22.42 |
| Proficiency and mastery | Degree of mastery 80% | 98.0% | |
| | Degree of proficiency 80% | | 70.0% |

**Table 12.** Summarized achieved results of the effectiveness of 3D virtual physics laboratory using SLOODLE.

| Assessment | Aspect | Results | |
| --- | --- | --- | --- |
| | | Black Ratios | Experimental Group |
| Effectiveness of 3D virtual physics laboratory using SLOODLE | Cognitive aspect | 1.2 | 1.75 |
| | Performance aspect | 1.2 | 1.68 |
| | Skill side | 1.2 | 1.69 |

*Effectiveness of 3D Virtual Physics Laboratory*

There are three aspects designed to evaluate the effectiveness of the 3D virtual physics laboratory. Although this learning system may be effective for people who do not have learning disabilities, we concentrate on students with learning disabilities. Students with the largest average scores (5) are completely satisfied with the education in the 3D virtual physics laboratory, which is enjoyable and beneficial. The following evaluation aspects are revealed:

(a) The technical aspect concentrates on how easy it is to use the 3D virtual physics laboratory and what issues and problems students encounter while inside a 3D virtual learning environment.

(b) The second aspect is concerned with interactions in it, describing the capabilities of the 3D virtual physics laboratory as very interesting because they are interacting directly with some difficulty with objects such as a solar model.

(c) The academic aspects are the third category. Most students state that they participate in the 3D virtual physics laboratory experience to discover a new method. The 3D virtual physics laboratory allows students to learn about physics in a funny and easy way, in addition to the flexibility they find when they use the 3D virtual physics laboratory to take the class without being in class due to factors like illness. From the perspective of students with learning disabilities, Table 13 shows the effectiveness of the 3D virtual physics laboratory.

Figure 4 represents students with learning disabilities who have responded to many of the questions used to assess the effectiveness of the 3D virtual physics laboratory.

Figure 4 presents the variation of educational benefits issues aspect in the function of a number of disabled students.

**X1:** I gained a better understanding of physics concepts. I worked on honing my operational skills. It proved more efficient to fulfill the need for the experimental course this way.

**X2:** I gained a better understanding of simple physics concepts, but I still need some work on some experiments. It proved more efficient to fulfill the need for the experimental course this way.

**X3:** I still did not understand some experiments.

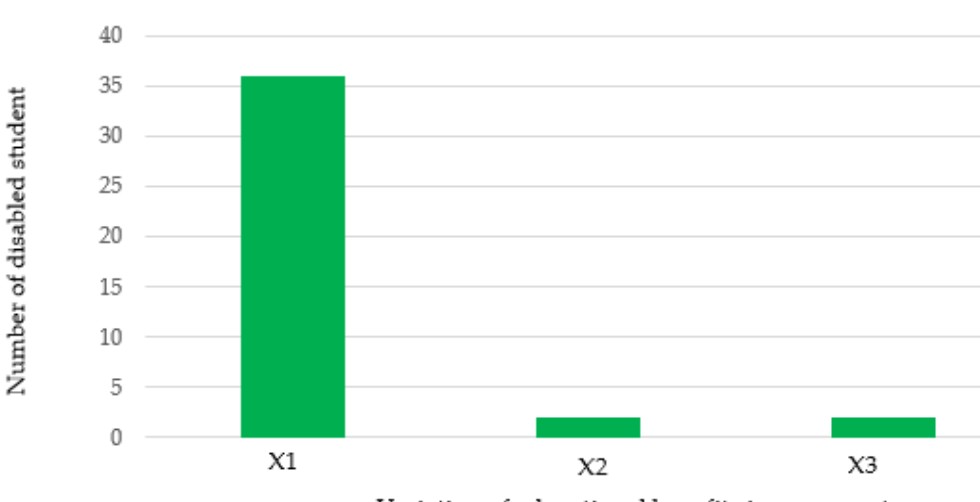

**Figure 4.** Academic Aspect of 3D virtual laboratory.

**Table 13.** Experience of a student with disabilities in 3D virtual physics laboratory.

| Aspects | Characteristics of Students' Experiences | | |
|---|---|---|---|
| | Easy | Medium | Difficult |
| Technical issues aspect | It is easy for me to interact with the 3D virtual physics lab, in addition to interacting with 3D virtual environment features like sound, flying around places, touching 3D objects and showing their physics features. I like the solar model as it teaches me a lot about planets. | While I can seek information in the lab with the aid of instructions. I often experience some anxiety when using the 3D physics virtual lab environment. I am confused when my destination changes. | I have a hard time using the 3D physics virtual lab. I am engaged since I have a difficult time figuring out how to conduct the experiment. |
| Interactive aspect | As I work in a 3D virtual physics lab, the interaction is simple and easy, and all the parts of the experiment are interesting to me, and I communicate with my partners throughout the experiment. | The interaction is simple and easy in basic experiments about shapes motion and reactions beyond fall, but I face some confusion when I communicate with my partners throughout the experiment. | I try harder to keep it all under control, but I can't. |
| Educational benefits (Academic aspect) | I gain a better understanding of the physics concepts. I work on honing my operational skills. It proves more efficient to fulfill the need for the experimental course this way. | I gain a better understanding of simple physics concepts, but I still need some work on some experiments. It proves more efficient to fulfill the need for the experimental course this way. | I still don't understand some experiments. |

Figure 5 shows the variation of interactive issues aspect for different disabled students.

**X1:** As I worked in a 3D virtual physics lab, the interaction was simple and easy, all the parts of the experiment were interesting to me, and I communicated with my partners throughout the experiment.
**X2:** The interaction was simple and easy in basic experiments about shapes motion and reactions beyond fall, but I faced some confusion when I communicated with my partners throughout the experiment.
**X3:** I tried harder to keep it all under control, but I cannot.

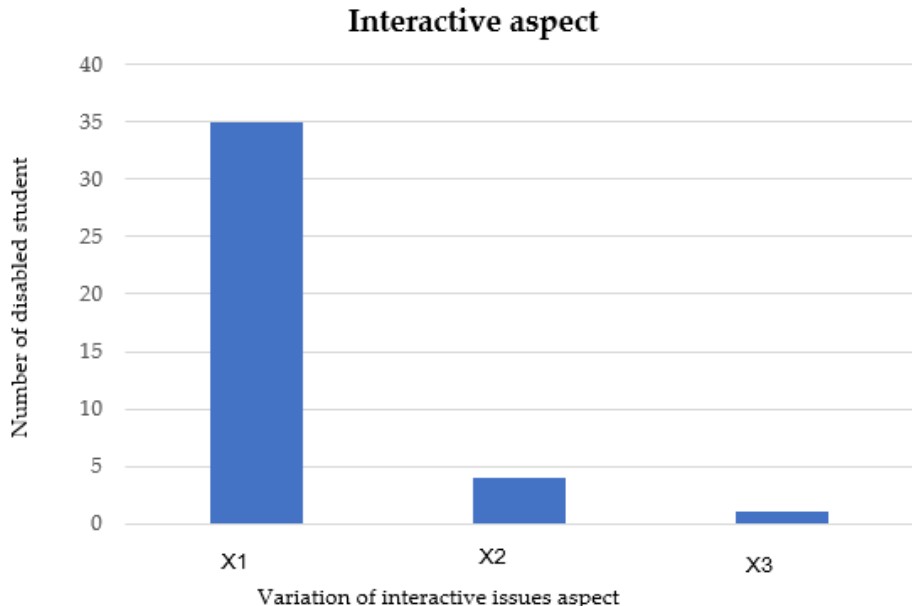

**Figure 5.** Interactive aspect of 3D virtual laboratory.

Figure 6 presents the technical issues of the 3D virtual laboratory of disabled students.

**X1:** It is easy for me to interact with the 3D virtual physics lab, in addition, to interacting with 3d virtual environment features like sound, flying around places, touching 3d objects, and showing their physics features, I like the solar model.

**X2:** While I can seek information in the Lab with the aid of the instructions, I often experienced some anxiety when using the 3D physics virtual lab environment.

**X3:** I have a hard time using the 3D physics virtual lab I engaged since I have a difficult time figuring out how to conduct the experiment.

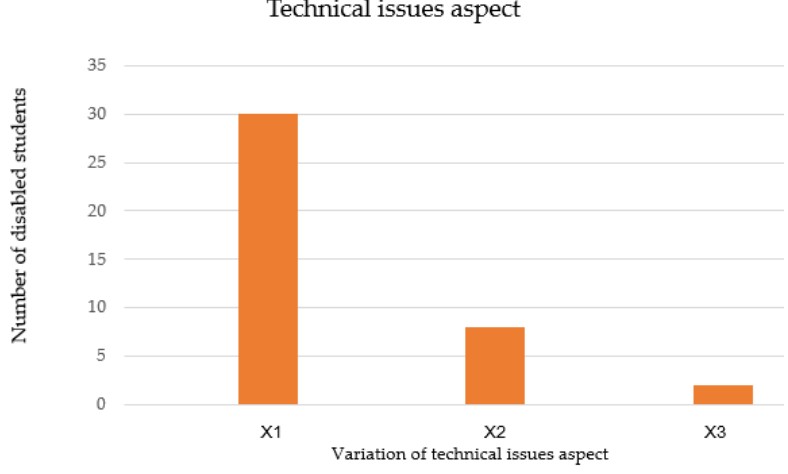

**Figure 6.** Technical issues of 3D virtual laboratory.

## 8. Conclusions

In this paper, we have developed and evaluated a 3D virtual physics laboratory to improve cognitive skills in physics experiments for students with learning disabilities. It has been created according to criteria specifically tailored for special needs students, as it includes a variety of three-dimensional objects of different colors that will attract the attention of students, besides various three-dimensional objects and models. Objects have been moved and reacted to falling in different shapes using simple physical experiments. The movement of the planet's rotation around their axis and other planets has been

simulated using a special model of the solar system that provides the necessary planets. Then, the planet's rotation and movement were simulated using the LSL script. The study focused on learning disabilities (dyslexia, dyscalculia, ADHD, and information retrieval disabilities) because those students need to conduct the physics of experiments. Depending on the nature of the developed system specifications, the system has provided students with learning disabilities with a tool and technique that integrates the emulator 3D VLE platform OpenSimulator to Moodle, designed to allow these students to obtain the 3D VLE in a standalone mode in a local network or by making the system available as an application on a large internal network or over the internet. The scheme proposed in this study could indeed conveniently manage and edit Moodle educational materials within the 3D VLE and interact with various disabled students. Through the email service, the 3D VLE connects the name of the user or virtual character in the OpenSim emulator to the same user in Moodle. Students with learning disabilities can perform a variety of tasks. Students with learning disabilities have improved their cognitive skills in physics through a 3D virtual physics lab environment with 15.50 for an experimental group compared with 11.15 for a control group, with 98% and 80% degree of mastery, respectively, and practical skills are 116.80 for the experimental group compared with 22.42 for the control group. Furthermore, according to Black's ratio of 1.2, the 3D virtual physics laboratory using SLOODLE has a score of 1.75 and 1.68 for the cognitive and performance aspects, respectively, and a score of 1.69 for the skill aspect.

**Author Contributions:** A.O.E. and S.S. contributed to the design and conception of the study. A.A.A. reorganized the paper and wrote the first draft of the manuscript. A.O.E. performed the statistical analysis. A.O.E. wrote the manuscript. S.S. and A.O.E. interpreted the data and analyzed the results. All authors have read and agreed to the published version of the manuscript.

**Funding:** The authors would like to thank the King Salman Center for Disability Research for funding this work through Research [Group no KSRG-2022-027].

**Data Availability Statement:** The dataset presented in this article is not readily available due to concerns regarding participant/patient anonymity. Requests to access the datasets should be directed to the corresponding author.

**Conflicts of Interest:** There are no issues related to the journal policies, nor are there any competing interests.

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
