# Peer review of "An Efficient System Based on Experimental Laboratory in 3D Virtual Environment for Students with Learning Disabilities"

_electronics, doi:10.3390/electronics12040989_

Round 1

Reviewer 1 Report

This paper explores the exciting topic of using virtual reality technology to enhance the education of persons with disabilities. The study describes a virtual physics laboratory used with a group of middle-school girls with disabilities in Saudi Arabia.

The system that was developed, and the physics concepts that it teaches, are not described in sufficient detail for the readers to ascertain the significance of the results of the experiments that are presented. The software used includes Moodle, which is well-known, and Sloodle, which is not well-known. Sloodle integrates Moodle into SecondLife, which is hardly mentioned in the paper, and is a very dated desktop virtual reality system at this point.

The English language in the paper is not bad but still has many issues.  For example, in several places the word "whose" is used incorrectly where the word "with" is needed instead.

"aims to develop and evaluate" should read "develops and evaluates"

"Students with learning disabilities can benefit from virtual reality" is casually asserted without evidence in the introduction, when it should instead be the main research question of the paper. More broadly, the research results presented are not clearly tied to open research questions for which they should be providing answers.

"interface. By giving" should read "interface, by giving"

"rather than allowing individuals" should just read "allowing individuals"

"and gain a sense" should read "and enable them to gain a sense"

"computer-simulated environment" should read "computer-generated environment", because the preceding verb is already "simulates"

"explore it, interact with it" should just read "explore, interact"

Various places in the paper have an extra space before a period that should not be there, or are missing a space after a period. All periods should be checked.

"child's time a task" makes no sense and should be rephrased

"who could advantage from learning. A job" should read "who could benefit from learning a job"

There are a lot of capital letters on words that are not beginning a sentence and are not proper nouns, so the word should not be capitalized.

"learning disabilities the following" makes no sense and should be rephrased

"pendency" is not a word in English.

"a usable design should be conforms to conventions" makes no sense.

"late 1990's, among" should read "late 1990's. Among"

"threedimensional" is used in multiple places but this is not a word. It should have a space or a hyphen between "three" and "dimensional"

The bulleted list in section 3 has many sentences that introduce a reference and are not ended in a period as they should, but instead end in a comma or a colon.

Section 4 talks about "this system" but has never properly named or described the system being discussed.

The sentence beginning "Make the system available" is not a sentence and should be rephrased.

"just only to" should be rephrased.

"the e-learning principle" is unclear and should be rephrased

"Server, and Server" does not make sense. Apache is a web server and PHP is a web technology not not exactly a server.

"installed, for" should be "installed. For"

bad linebreak later in that sentence in 5.1 d) before "registering virtual characters"

"Figure 3 present" should be "Figure 3 presents"

"perimental" is not a word

"we had been using" should read "we used"

"ranyah - taif" is lacking capitalization and is phrased differently elsewhere.

"hypothe" is not a word

In Table 2, bad linebreak in "behavioral goal levels"; should linebreak between words.

In Table 2, Summation line sums to 100% but the units in other columns do not seem to be percentages or add up to 100.

"We also determined...to determine" does not make sense and uses "determine" too much.

numbered tables are proper names and should read, e.g. "Table 6" instead of "the table 6"

In discussion of statistics, commas are used european style where decimals are used in the tables being quoted. A consistent style for real numbers should be used.

,)102(  is badly formatted

"are average They suffer" makes no sense and surrounding text should be phrased into legal sentences.

"the3 D" should read "the 3D"

"Table(10)" should read "Table 10"

"models on which have practice" makes no sense and should be rephrased

chat is "largely seems to have a spurious doublequote

Reference #10 seems to be garbled and mixes two or more references

Author Response

Response to Reviewer 1 Comments

Response 1: The system that was developed, and the physics concepts that it teaches, are not described in sufficient detail for the readers to ascertain the significance of the results of the experiments that are presented

3D virtual physic laboratory experiments:

As part of our training, we examined the movement of simple shapes such as balls, cones, and cubes in three dimensions, along with Newton's three laws, along with one of Kepler's laws, along with how their fall is affected by the shape. Various areas were carved out of the virtual laboratory's three dimensions, each of them consists of specific experiments, such as an area where planetary motion laws were tested and experimented with, Using the LSL script, special effects were programmed by adding details about the planets to the script. As shown below, these forms contain an explanation and detail of Newton and Kepler's laws:

            Kepler's laws of planetary motion [21]

In the beginning, there is:

All planets orbit the Sun in the form of an ellipse Mathematically, an ellipse can be represented by the formula:

r="ρ" /(1+ε cos⁡θ )

            Newton's first law of motion of a body [22]

Unless compelled to change its state by forces impressed upon it, everybody remains at rest or in uniform motion in a straight line.

            Newton's second law of body motion (Circular motion) [22]

Second, an object's motion is proportional to the force impressed and occurs in the direction of the straight line in which the force is impressed.

            Newton's third law of body motion [22]

There is always an equal reaction to every action; or the mutual actions of two bodies on each other are always equal and directed to opposite parts.

Point 2 : The English language in the paper is not bad but still has many issues.  For example, in several places the word "whose" is used incorrectly where the word "with" is needed instead.

done in each place appears (students with disabilites)

Point 3: "aims to develop and evaluate" should read "develops and evaluates"

develops and evaluates

point 4 : "Students with learning disabilities can benefit from virtual reality" is casually asserted without evidence in the introduction, when it should instead be the main research question of the paper. More broadly, the research results presented are not clearly tied to open research questions for which they should be providing answers.

Discussion of achieved results

Our proposed 3D virtual physics laboratory was developed using simulation model (Open Simulator) repository files that are installed through the simulation program's file system and therefore are files with special extensions (Oar,iar), and all the basic functions associated to the 3D virtual environmental management systems (Sloodle) have been installed, such as student registration in Learning management system (Moodle) directly from the 3D virtual environment. Those students who used the 3D virtual physics lab achieved high levels of achievement in the study, demonstrating the effectiveness of the virtual reality lab for students with learning disabilities, along with the demonstration of the effectiveness of SLOODLE 3D virtual physics laboratory is shown in table 11 and table 12.:

Table 11. summarized achieved results of Cognitive assessment of the 3D virtual physics lab:

Assessment

Aspect

Results

T-test (α=0.05)

Experimental group's

Control group's

Cognitive assessment of practical experience skills

cognitive test

13.838

15.50

11.15

Performance

18.7

101.30

74.40

Practical skills

85.55

116.80

22.42

Degree of mastery (80%)

36.075

98.0%

Degree of proficiency (80%)

9.63

70.0%

Table 12. summarized achieved results of the effectiveness of the3d virtual physics laboratory using SLOODLE:

Assessment

Aspect

Results

Black adjusted gain ratios

Experimental group's

The effectiveness of the3d virtual physics laboratory using SLOODLE

Cognitive aspect

1.2

1.75

Performance aspect

1.2

1.68

The skill side

1.2

1.69

Point 5: "interface. By giving" should read "interface, by giving"

interface, by giving

point 6: "rather than allowing individuals" should just read "allowing individuals"

allowing individuals

point 7: "and gain a sense" should read "and enable them to gain a sense"

and enable them to gain a sense

point 8: "computer-simulated environment" should read "computer-generated environment", because the preceding verb is already "simulates"

computer-generated environment

point 9 :"explore it, interact with it" should just read "explore, interact"

explore, interact

point 10: Various places in the paper have an extra space before a period that should not be there, or are missing a space after a period. All periods should be checked.

Done

Point 11: "child's time a task" makes no sense and should be rephrased

student’s time a task

point 12: "who could advantage from learning. A job" should read "who could benefit from learning a job"

who could benefit from learning a job

point 13: There are a lot of capital letters on words that are not beginning a sentence and are not proper nouns, so the word should not be capitalized.

done

point 14: pendency" is not a word in English.

  1. c) Using attractive colours will catch the child's attention and reduce his or her tendency to be violent

point 15 : a usable design should be conforms to conventions" makes no sense

a usable design should be

point 16: "late 1990's, among" should read "late 1990's. Among"

late 1990's. Among

point 17 : "threedimensional" is used in multiple places but this is not a word. It should have a space or a hyphen between "three" and "dimensional"

three dimensional

point 18 : The bulleted list in section 3 has many sentences that introduce a reference and are not ended in a period as they should, but instead end in a comma or a colon.

Done

Point 19 : Section 4 talks about "this system" but has never properly named or described the system being discussed.

The design goal of this system (3D virtual physics laboratory) is to establish an interactive communication platform for teachers and students with learning disabilities, particularly in physics experiments, so the authors proposed to develop and evaluate a 3D virtual Reality Laboratory to improve cognitive skills in Physics experiments for students with learning disabilities. Depending on the scope of the below requirements, the system must provide us with a tool and technology to integrate the emulator of the  3D virtual environment  (Open Sim) that was used to emulate the 3d virtual laboratory and then connect it with the learning management system (Moodle) and manages the latter from inside the 3D virtual environment of the 3D physics laboratory that means we used open-source software including OpenSim as well as LMS (MOODLE) which is integrated with Open-Sim through SLOODLE technology, Furthermore, the Sloodle project which is an open source project that aims to integrate web-based learning features. which allows us to bring a virtual Reality Laboratory to the 3D virtual environment available on the web. The system will be available as an application on a large internal network or over the web. [20].

Our proposed 3D virtual physics laboratory was developed using simulation model (Open Simulator) repository files that are installed through the simulation program's file system and therefore are files with special extensions (Oar,iar), and all the basic functions associated to the 3D virtual environmental management systems (Sloodle) have been installed, such as student registration in Learning management system (Moodle) directly from the 3D virtual environment.

Point 20: The sentence beginning "Make the system available" is not a sentence and should be rephrased.

The system will be available as an application on a large internal network or over the web

Point 21: "just only to" should be rephrased.

  • The user of the 3D virtual physical laboratory is not allowed to modify the content of the 3D virtual environment, just interact with it.

"the e-learning principle" is unclear and should be rephrased which is the e-learning method

"Server, and Server" does not make sense. Apache is a web server and PHP is a web technology not not exactly a server.

  • Wamp Server: This server was chosen because it is open-source for the Windows operating system and hosts the applications required to run the technologies. These include Databases (MYSQL), Programming Language (PHP), and Server (Apache), and it creates a local server on your machine that hosts the database application (MYSQL), the 3D virtual environment (OpenSim), and the learning management system databases (Moodle).

"installed, for" should be "installed. For" installed. For

bad linebreak later in that sentence in 5.1 d) before "registering virtual characters"

For instance, we select enrollment booth to assist with the procedure of registering virtual characters in the learning management system (Moodle)

"Figure 3 present" should be "Figure 3 presents" Figure 3 presents

"perimental" is not a word

"we had been using" should read "we used" we used

"ranyah - taif" is lacking capitalization and is phrased differently elsewhere TAIF – RANYAH

In Table 2, bad linebreak in "behavioral goal levels"; should linebreak between words. done

In Table 2, Summation line sums to 100% but the units in other columns do not seem to be percentages or add up to 100. done

"We also determined...to determine" does not make sense and uses "determine" too much. Done

Point 22:numbered tables are proper names and should read, e.g. "Table 6" instead of "the table 6" e.g table

Point 23 : In discussion of statistics, commas are used european style where decimals are used in the tables being quoted. A consistent style for real numbers should be used.

36.075  9.631  22.422 18.788 etc

Point 24: ,)102(  is badly formatted (102)

Point 25: "the3 D" should read "the 3D" 3D

"Table(10)" should read "Table 10" Table 10

"models on which have practice" makes no sense and should be rephrased

as shown in Figure severalties lab contained a set of 3D models which can practice  with it a number of basic experiments were applied

chat is "largely seems to have a spurious doublequote done

point 26: Reference #10 seems to be garbled and mixes two or more references fixed it

Reviewer 2 Report

 In this paper, you developed and evaluated a 3D virtual physics laboratory to improve cognitive skills in physics experiments for students with learning disabilities.

 The objectives and approach of the proposed system are well understood, but the paper is insufficient for acceptance with respect to the following points and should be revised.

 The information provided in 1. Introduction gives the impression that it is the opinion of the authors. Please cite the literature properly and explain logically based on evidence.

 I did not understand the advantage of the proposed method over the methods of [14][15]. Please clarify these.

 Experimental reproducibility and reliability are lacking in the experiments to demonstrate the usefulness of the proposed method. Please revise your paper according to the following remarks.

- You have not described what "the conventional process" is used in the experiment to compare the proposed system. Please describe exactly what "the conventional process" is, as the results can be produced in any way depending on the method.

- Show that the learners, 20 randomly divided learners each, have similar learning ability with respect to physics before the experiment.

- Although we recognize that the proposed method is a system specifically for people with learning disabilities, it is possible that this learning system is also effective for people who do not have learning disabilities. In other words, it is possible that the good results are due to the quality of the learning content. Please provide the results of your experiments that show what aspects of the proposed system are effective for people with learning disabilities.

 Although you state "The environment was created based on specially designed criteria for disabled students.", no specific content is described, so it is not reproducible by the reader. A part of the actual implemented contents are specifically described in the chapter 7. Discussion of achieved results, but these should be specifically explained using Figure 5 in the 5. implementation section.

 Due to the low resolution of the figure, some text in the figure is unreadable. Please correct the figure.

 There are many typographical errors and areas that do not follow the specified paper format. Carefully review and correct the full text.

Author Response

Response to Reviewer 2 Comments

Point 1: The information provided in 1. Introduction gives the impression that it is the opinion of the authors. Please cite the literature properly and explain logically based on evidence.

Response 1: Methodology of the proposed Approach

The design goal of this system is to establish an interactive communication platform for teachers and students with learning disabilities, particularly in physics experi-ments, so the authors proposed to develop and evaluate a 3D virtual Reality Labora-tory to improve cognitive skills in Physics experiments for students with learning disa-bilities. Depending on the scope of the below requirements, the system must provide us with a tool and technology to integrate the emulator of the  3D virtual environment  (Open Sim) that was used to emulate the 3d virtual laboratory and then connect it with the learning management system (Moodle) and manages the latter from inside the 3D virtual environment of the 3D physics laboratory that means we used open-source software including OpenSim as well as LMS (MOODLE) which is inte-grated with Open-Sim through SLOODLE technology, Furthermore, the Sloodle project which is an open source project that aims to integrate web-based learning features. which allows us to bring a virtual Reality Laboratory to the 3D virtual environment available on the web. The system will be available as an application on a large internal network or over the web. [20].

Point 2: Experimental reproducibility and reliability are lacking in the experiments to demonstrate the usefulness of the proposed method. Please revise your paper according to the following remarks.

- You have not described what "the conventional process" is used in the experiment to compare the proposed system. Please describe exactly what "the conventional process" is, as the results can be produced in any way depending on the method.

Response 2: The conventional process consists of various physics experimental displays such as video about planets motion and rotation as well as some 3d models like cube, a ball, a cone etc used to execute practical experimental physics inside lab.

Point 3. - Show that the learners, 20 randomly divided learners each, have similar learning ability with respect to physics before the experiment.

Students with disabilities were divided into two groups based on their disability type. Students with attention deficits, for example, were given experiments on three-dimensional objects with appealing colors, and students with poor concentration and difficulty with mathematical operations were given planetary rotation motion experiments in the solar system in addition to motion experiments. For various bodies, as reactions based on the body's initial starting speed until it stops, with the provision of functions that calculate the body's speed based on its mass.

Point 4  - Although we recognize that the proposed method is a system specifically for people with learning disabilities, it is possible that this learning system is also effective for people who do not have learning disabilities. In other words, it is possible that the good results are due to the quality of the learning content. Please provide the results of your experiments that show what aspects of the proposed system are effective for people with learning disabilities.

To assess effectiveness there were three aspects designed to evaluate the effectiveness of the Virtual 3D physics laboratory. Although this learning system may be effective for people who do not have learning disabilities, we concentrated on students with learning disabilities. The highest median scores (5) indicated that students were fully satisfied with the education in the Virtual 3D physics laboratory which was enjoyable and beneficial. The following evaluation aspects are revealed:

  1. The technical aspect concentrated on how easy it is to use the Virtual 3D Physics Laboratory as well as what issues and problems students encountered while they were inside a 3d virtual learning environment.
  2. The second aspect dealt with interactions. Students interested in increasing their engagement with their teachers described the capabilities of the Virtual 3D physics laboratory to assist interactions and communication which was very interesting because they were interacting directly with some objects were difficult to interact with it, for instance, a solar model, aside from the educational use evaluated in the study.
  3. The academic aspects were the third category. Most students stated that they participated in the Virtual 3D physics laboratory experience to discover a new method. Virtual 3D physics laboratory allows students to learn about physics in a fun and easy way. In addition to the flexibility that they found when they used Virtual 3d physics laboratory to take the class without being in class due to factors like illness. From the perspective of students with disabilities, the following table 13 shows the effectiveness of the Virtual 3D Physics Laboratory.

Table 13. The experience of a student with disabilities in the Virtual 3D Physics Laboratory

Aspects

Students’ descriptions of experience

easy

medium

difficult

Technical issues aspect

It is easy for me to interact with the 3D virtual physics lab, in addition, to interacting with 3d virtual environment features like sound, flying around places, touching 3d objects, and showing their physics features, I like the solar model it taught me a lot about planets

While I can seek information in the Lab with the aid of the instructions, I often experienced some anxiety when using the 3D physics virtual lab environment.

I am confused when my destination changes"

I have a hard time using the 3D physics virtual lab I engaged since I have a difficult time figuring out how to conduct the experiment

Interactive aspect

As I worked in a 3D virtual physics lab, the interaction was simple and easy, and all the parts of the experiment were interesting to me, and I communicated with my partners throughout the experiment.

the interaction was simple and easy in basic experiments about shapes motion and reactions beyond fall, but I faced some confusion when I communicated with my partners throughout the experiment.

I tried harder to keep it all under control, but I can't

Educational benefits (Academic aspect)

I gained a better understanding of the physics concepts. I worked on honing my operational skills. It proved more efficient to fulfill the need for the experimental course this way.

I gained a better understanding of simple physics concepts, but I still need some work on some experiments.  It proved more efficient to fulfill the need for the experimental course this way.

I still didn’t understand some experiments

Also figure 4 represents students with disabilities responsables of many questions to evaluate the effectiveness of the Virtual 3D Physics Laboratory.

Figure 4. The effectiveness of the Virtual 3D Physics Laboratory

Point 5 Although you state "The environment was created based on specially designed criteria for disabled students.", no specific content is described, so it is not reproducible by the reader. A part of the actual implemented contents are specifically described in the chapter 7. Discussion of achieved results, but these should be specifically explained using Figure 5 in the 5. implementation section.

A certain attribute of a 3D virtual physics laboratory is considered a particular criterion for enabling students with disabilities to learn with special content, the criteria included the color for the(balls, cones, cubes, etc.) which were 3D models, the way 3D models fall in accordance with physics laws, the interactive with sound that draws students' attention, furthermore fly feature which enable user fly around 3D virtual laboratory especially when the matter about knowing student with learning disabilities about planets rotation and their motion. Below summarized these criteria:

  1. Using attractive colors will catch the child's attention and reduce his or her tendency to be violent of written information.
  2. Using notecards to demonstrate experiments steps and explain it.

iii.           Students with learning disabilities can display presenter tool allow student show whole physics experiment as PDF files, web pages, and videos. 

  1. Students with learning disabilities can use glossary tool which they can use this tool to identify the terms in Moodle. To obtain immediate results, stu-dents or teachers must enter the word "/def" followed by the name of the term for which they want to search, which is registered in the glossary.

Point 6: Due to the low resolution of the figure, some text in the figure is unreadable. Please correct the figure.

Point 7 : There are many typographical errors and areas that do not follow the specified paper format. Carefully review and correct the full text. Done

Round 2

Reviewer 1 Report

This paper presents a virtual 3D physics laboratory allowing users to experiment with a few of Newton's Laws at the elementary or perhaps middle school level. The software system is mainly an assembly of pre-existing tools (OpenSim+Moodle via Sloodle). The interesting results presented show that students with disabilities performance improved when using the software when compared with not using the software.

A very basic description of the physics being taught was added to the paper, but it is still not very clear the extent of the content that was learned or the extent of the accuracy of the physics depicted in the virtual world. For example, were objects being pulled by gravity depicted in a vacuum, or falling through air? Are the presented research results depicting student performance improvement on a single one-hour lesson, on a week's worth of physics topics, or on a larger set of knowledge?

The paper has many English language issues that impede reader understanding. Some of them are given below, but there are probably too many for the reviewer to catch in one reading.

Abstract:

"(Dyslexia, Dyscalculia, attention issues (ADHD), and disability in retrieving information)" should not have the outer parentheses and instead should read "dyslexia, dyscalculia, attention issues (ADHD), and disability in retrieving information"

"which students need to do physics experiments" is a clause that needs commas before and after ", which students need to do physics experiments, "

Introduction:

It does not make sense to define the acronym VLE from the phrase "Virtual classroom environments". You should either use "Virtual learning environments" to match VLE, or change the acronym to VCE if you think that is widely used.

Saying that a virtual classroom environment is "known as" (i.e. a synonym for) a course management system seems to be a false equivalence.  The two concepts are related, but not identical.

There are at least two (possibly more) major software packages called OpenSim. The authors should find an appropriate primary source to cite, or at least give the website for OpenSim to disambiguate the tool they are using. Since LSL is a SecondLife thing, and SLoodle was seemingly written originally for SecondLife, it is worth mentioning whether the 3D Physics Laboratory could be just as easily implemented in SecondLife, or whether OpenSim has compelling technical or legal advantages that make it preferable.

"On other hand" should read "On the other hand"

".[2] ."  should be "[2]."

"The classification of the students with disabilities are ten" should read "The classification of the students with disabilities consists of ten" -- and whose classification is this? Who says there are (only) ten categories?

"ASD [3]" is missing a period and should read "ASD [3]"

"So, students with learning disabilities can benefit from virtual reality (VR)." is a claim that has not been proven at this point in the paper. Perhaps it is an axiom or an opinion, or perhaps it is a research hypothesis that you are yourselves trying to prove. It should be rephrased as one or the other: an axiom or a hypothesis.  This could be as possible as changing "So, " to "We believe", but the paper would be stronger if you start a new paragraph here and state this as a hypothesis.

"abilities.[2]" should read "abilities [2]."

"mobility.[4]" should read "mobility [4]."

"participate.[4]" should read "participate [4]."

"mastery.[5]" should read "mastery [5]."

"2. Literature view" should perhaps read "2. Literature Review". But what is the difference between "2. Literature view" and "3. Related Work"? Section 2. should be renamed to clarify, if it is not part of the related work section.

"(audio/video)" serves no purpose immediately after "audio and video input" and should be deleted. Similarly, how are "speakers and/or headphones" anything other than audio input already mentioned?  HMD's may indeed have additional sensory input besides audio and video -- rephrase this sentence.

In 2.2 Personalizing VR:

"hard else or impossible for them" should read "difficult or impossible for them otherwise"

"student's time a task" has a grammar problem, it should read "student's time to complete a task" or "student's time on task"

"significant effect on the learner's special ability" is unclear.  What special ability?

"Special education" is inconsistent. Probably should say "Special Education"

"when designing and production" is inconsistent. Should say "when designing and producing" or possibly "during the design and production"

"a) Determine ... b) Ease of use. c) Analyzing..." is inconsistent. Should rephrase so a) - c) are all verb phrases of the same type, or all noun phrases.

"a) Preferred... b) It is preferable" is inconsistent. The a) part should be rephrased as it is not a sentence.

"and demonstrate as the following" is unclear and should be rephrased.

"to be violent of written communication" is unclear and should be rephrased.

It occurs twice, in the related work and in your own system requirements.

"be as the bellow" should be rephrased, perhaps just "be:"

"a) easy... b) conforms... c) adaptable" is inconsistent, with a) and c) being adjectives but b) being a verb.

In Section 3. Related Work:

"this major Since" should read "this topic since"

"This paper discusses A desktop" should read "In this paper, a desktop"

", [16] This" should read " [16]. This"

"presents the research examined" should read "presents research that examines"

"skills,[17]" should read "skills [17]."

"disabilities [18] This" should read "disabilities [18]. This"

The discussion of [18] is at present too vague to be useful to this paper.

"Needs [19]:" uses all capitalized words for the title of the paper, which is inconsistent with the format of the preceding papers. It uses a colon here, which should either be a period, or all preceding papers should use colon.

In section 4. Methodology:

Capitalization in "Methodology of the proposed Approach" is inconsistent.

The sentence beginning "The design goal of this system" is awkward and should be rephrased and split into multiple sentences.  A design goal is a different sort of thing than is "establish an interactive communication platform", which sounds more like an overall goal or an implementation goal.

The sentence beginning "Depending on the scope" is a long run-on sentence that should be broken up into multiple smaller sentences that can be understood.

For the long list of system requirements, it is unclear how much of this is already provided by OpenSim or Sloodle, versus how much you had to invent.

Maybe subdivide requirements into two categories, those that are provided by the underlying off-the-shelf components and those that required development or creative solutions on your part.

Parentheses should be removed in "(inviting friends... Etc.)"

", so to enable" should read ". To enable"

"the(balls, cones, cubes, etc.)" should read "the primitive shapes (balls, cones, cubes, etc.)"

"the interactive with sound" is unclear and may be missing a word.

"interactive" is an adjective so it requires a following noun.

"attention, furthermore fly feature which enable user fly around" should read "attention. Furthermore, the flying feature, which enables the user to fly around the"

"especially when the matter about knowing" is unclear and should be rephrased

"glossary tool which they can use this tool" is unclear and should be rephrased

"architectural style" should be "architecture"

"(Sloodle) software" should be "Sloodle software"

"learning management system (Moodle)" can just be "learning management system"

In Table 1 there are awkward linebreaks in the 2nd and 5th rows that harm readability.  Probably the width of column 1 can be reduced to make column 2 wider and avoid these awkward linebreaks. Alternatively, you could insert linebreaks (after "level" and "capabilities") in order to avoid the bad ones

The phrase "learning management software, which is the e-learning method" does not make sense as-written. It should be corrected or clarified.

In Figure 1, "cha" should be "chat" in the right column

In Section 5. Implementation:

"Figure severalties lab" makes no sense and may be missing words.

"practice with it a number" makes no sense and may be missing words or punctuation.

The sentence containing "speed of falling, a number of 3D models ..." makes no sense and should be split into multiple sentences. Within it, the phrase "(cube - A ball - a cone)" should read "such as a cube, a ball, and a cone". That phrase has grammar problems; perhaps a verb is missing after it.

The sentence containing "The students also liked..." seems overlong and should be split into multiple sentences.

The two sentences "The students also used..." followed by "The students with learning disabilities also used..." seem redundant and should be consolidated.

"the following stages were implemented in the system according to the following" should read "the system was implemented in the following stages"

"Establishing the virtual lab building in this phase, a lab building" should read "Establishing the virtual lab building. In this phase, a lab building"

"presentations, we put" should read "presentations. We put"

The tense used in "would have to become..." and "the two were becoming..." is awkward. Use a more clean past tense to describe how Sloodle is used to integrate tools.

"the PowerPoint presentation that Established in Moodle chatting tool" makes no sense and should be rephrased.

"fully prepared to use" should read "fully prepared for use" or "fully prepared to be used"

In Section 5.2:

"physic laboratory" should read "physics laboratory".  Also, the subsection title should not end with a colon, and should be formatted consistently with Section 5.1, perhaps in italics.

"experimented with," should read "experiments conducted."

"ellipse Mathematically" should read "ellipse. Mathematically"

"everybody" should read "every body"

In Section 6:

"which they have" should read "that had"

"students, Students" should read "students. Students"

"table 2 below" should read "Table 2 below"; because Table 2 is a proper noun it should be capitalized.

"fifth hypotheses" should read "five hypotheses"

The five hypotheses are all missing a required "The" before "first", "second", etc.  Alternately, the phrases "First hypothesis is" through "Fifth hypothesis is" can all be deleted.

"(traditional method)" should read "traditional method". This occurs in multiple places. It feels like "(traditional method)" was from some template language that is being copied in here and the authors were supposed to replace it with the method that the control group used.

"e.g table 3" should read "e.g. Table 3". Same correction later for "e.g table 5", "e.g table 6", "e.g table 8", "e.g table 9", "e.g table 11".

The names of particular tables are proper nouns and are to be capitalized.

Table 3 contradicts the text that describes it. The text claims that the experimental group performed better when compared to the control group, but the numbers in the table are reversed from the numbers reported in the text.

"In favor of the 424 experimental group which has a higher average." is not a sentence.

The phrase "We also calculating" needs to be rephrased in correct English.

Table 5 caption is on a different page from the table; they should be kept together on the same page.

Table formatting seems to place adequate space between body text and the start of a table, but no space between the end of the table and the following body text. Some lines on tables have unusual marks above or below the line.

"was ,00000" seems to have an extra comma and maybe a missing period

Odd break and missing capitalization on line starting "we used the t-test"

Odd use of parentheses around "(80)%", "(70%)", "(80%)". Remove paretheses.

"Black, (1.2)" should read "Black (1.2)." Start a following new sentence with "Therefore"

"Taif. the second" should read "Taif. The second". Similar fix later for "Taif. the third"

Odd break and non-sentence starting "3D virtual physics laboratory in developing" is unclear. Is this the start of a subsection? It needs formatting and/or grammar correction.

The phrase 'for" Black .The' should read 'for Black. The'

"the3D" should read "the 3D" (multiple occurrences)

"sloodle In" should be "sloodle. In".  Should sloodle be capitalized everywhere?

Similarly, SLOODLE should probably be capitalized Sloodle to be consistent with other uses (multiple occurrences).  Actually sometimes it is referred to as SLoodle.  What spelling did its creators use?  Use that spelling everywhere.

"in table 10" should read "in Table 10"

In Section 7. Discussion of achieved results

Poor linebreaks in Table 11 ("as-" and "ex-". Make that column a little wider.

Table 12 caption is on a separate page from Table 12. Keep them on the same page.

In Section 7.1 The effectiveness of the 3D virtual physics Laboratory, "Laboratory" should not be capitalized when nothing else is. Check journal required format and capitalize section titles as directed.

"To assess the effectiveness there" should read "There". The first four words add nothing and should be deleted.

Delete the period in "satisfied. with".

"3d" in item A. should probably be capitalized consistent with other uses: "3D"

All other occurrences of "3d" should also be fixed.

Item B contains a long run-on sentence. Split it into multiple sentences.

Sometimes "3D Virtual Physics Laboratory" is all capitals and other times it is "Virtual 3D physics laboratory". The name should be made consistent (both capitalization and whether it is 3D Virtual or Virtual 3D) across all the many uses of this phrase.

"table 13 shows" should read "Table 13 shows"

"figure 4 represents" should read "Figure 4 represents"

"represents students with learning disabilities answer of many" does not make sense and should be rephrased.

Figure 4 omits much detail and does not show enough information for the reader to draw any conclusions about the student feedback. The supporting body text should elaborate and make clear the language of the various rows for the student choices.

In Section 8. Conclusion:

"learning disabilities... that students need to conduct physics experiments" is unclear. Are you saying that students need disabilities to conduct physics experiments?

"the system should provide" seems like a future tense. In a conclusions section you are mostly talking past tense about the research that you did.

In the References:

Reference 2 link/URI should be simplified.  Reference 2 should give the author and date since the article lists those.

Reference 4 should include publisher and page numbers.

Reference 6 article title capitalization is inconsistent with other references.

Reference 8 missing space after 257-277

Reference 9 capitalization of "learner's" is inconsistent with rest of title.

Reference 13 is missing page numbers.

Various references are missing their publisher.

Reference 21 "Newten" is "Newton". "avaliable" should read "available". Font is different from other references.

References 21-22: Wikipedia is not a great academic reference. I am not sure you need references for Newton's and Kepler's laws, but if you need to cite something for them, cite work written by Newton and Kepler.

Author Response

Response to Reviewer 1 round2 Comments

Response 1: A very basic description of the physics being taught was added to the paper, but it is still not very clear the extent of the content that was learned or the extent of the accuracy of the physics depicted in the virtual world. For example, were objects being pulled by gravity depicted in a vacuum, or falling through air? Are the presented research results depicting student performance improvement on a single one-hour lesson, on a week's worth of physics topics, or on a larger set of knowledge?

The virtual physics lab dealt with a number of experiments related to the fall of objects from various heights in the air, noting the speed of the reactions of the fall depending on the shape and weight of the body according to Newton's laws of motion, through the possibility of the student interacting with these bodies of different shapes if he can by touching The body can interact with it by touch and increase its size and weight to test the movement of its fall with the difference of this size and weight. These are three physical laws that establish the science of body movement, and these laws link the forces affecting the body's movement, as well as an experiment related to the movement of planetary rotation, because the orbit of each planet is an ellipse, and the sun is located in one of its foci, as described by Kepler's first law of planetary motion. Close movement, in which the student can be present near these planets and interact with them according to the solar system model, which allows displaying information about the planets, their rotation speed, and their distance from the sun and the rest Planets, by interacting with them, this information appears as soon as the student approaches the planet. These experiments were carried out over the course of two weeks at the end of the semester, with two lessons per week lasting 40 minutes. During this time, pre-tests were also carried out in which experiments were carried out without the use of a Physics laboratory. The virtual classroom through explanations and watching videos about these experiments, then a pre-test for the students on these experiments and after, in which the experiments were conducted using the virtual physics laboratory, and then a post-test on these experiments, as the results showed an improvement in the students' performance in Post-tests compared to pre-tests.

Point 2 : The paper has many English language issues that impede reader understanding. Some of them are given below, but there are probably too many for the reviewer to catch in one reading.

"(Dyslexia, Dyscalculia, attention issues (ADHD), and disability in retrieving information)" should not have the outer parentheses and instead should read "dyslexia, dyscalculia, attention issues (ADHD), and disability in retrieving information"

"dyslexia, dyscalculia, attention issues (ADHD), and disability in retrieving information"

"which students need to do physics experiments" is a clause that needs commas before and after ", which students need to do physics experiments, "

, which students need to do physics experiments,

Introduction:

It does not make sense to define the acronym VLE from the phrase "Virtual classroom environments". You should either use "Virtual learning environments" to match VLE, or change the acronym to VCE if you think that is widely used.

Virtual learning environments

Saying that a virtual classroom environment is "known as" (i.e. a synonym for) a course management system seems to be a false equivalence.  The two concepts are related, but not identical.

Virtual learning environments (VLEs), is learning tools that incorporate computers and the Internet into instruction to enhance a student's learning experience Further-more they are increasingly being used in a variety of settings, including classrooms, informal learning environments, distance learning, business, and many others.

Point 3: There are at least two (possibly more) major software packages called OpenSim. The authors should find an appropriate primary source to cite, or at least give the website for OpenSim to disambiguate the tool they are using. Since LSL is a SecondLife thing, and SLoodle was seemingly written originally for SecondLife, it is worth mentioning whether the 3D Physics Laboratory could be just as easily implemented in SecondLife, or whether OpenSim has compelling technical or legal advantages that make it preferable.

Open Simulator server: it was chosen because it is an open-source application. It is a server that allows us to create and design and emulate a 3D virtual environment to which the 3D virtual environment browser is connected to display the 3D vir-tual environment.OpenSimulator can be used to create virtual worlds similar to Second LifeTM. OpenSimulator, on the other hand, does not intend to be a clone of the Second Life server platform. Rather, the project's goal is to enable the development of innovative features for virtual environments and the Metaverse as a whole.

Features

Supports online, multi-user 3D environments with as few as one simulator or as many as thousands.

Supports variable-size 3D virtual spaces.

Supports multiple clients and protocols - access the same world via multiple protocols at the same time.

Real-time, simulator-side physics simulation is supported.

Clients who create 3D content in real time are supported.

LSL/OSSL in-world scripting is supported.

Allows for unlimited customization of virtual world applications through the use of scene modules.[24][25]

"On other hand" should read "On the other hand" done

".[2] ."  should be "[2]." done

"The classification of the students with disabilities are ten" should read "The classification of the students with disabilities consists of ten" -- and whose classification is this? Who says there are (only) ten categories?

according to American Psychiatric Association the classification of the students with disabilities are ten categories: intellectual disabilities, autism spectrum disorder (ASD), attention-deficit/hyperactivity disorder, global developmental delay, communication disorders, specific learning disorders, developmental coordination disorders, stereo-typical movement disorders, tic disorders, and Tourette’s syndrome. Among these dis-orders, the highest rates are seen for intellectual disability and ASD

"ASD [3]" is missing a period and should read "ASD [3]"

point 4 : "Students with learning disabilities can benefit from virtual reality" is casually asserted without evidence in the introduction, when it should instead be the main research question of the paper. More broadly, the research results presented are not clearly tied to open research questions for which they should be providing answers.

In our paper, we are therefore attempting to prove that virtual reality can benefit students with learning disabilities

"abilities.[2]" should read "abilities [2]." done

"mobility.[4]" should read "mobility [4]." done

"participate.[4]" should read "participate [4]." done

"mastery.[5]" should read "mastery [5]." done

Point 4 "2. Literature view" should perhaps read "2. Literature Review". But what is the difference between "2. Literature view" and "3. Related Work"? Section 2. should be renamed to clarify if it is not part of the related work section. Done Literature Review

Point 5 "(audio/video)" serves no purpose immediately after "audio and video input" and should be deleted. Similarly, how are "speakers and/or headphones" anything other than audio input already mentioned?  HMD's may indeed have additional sensory input besides audio and video -- rephrase this sentence.

 in addition to the virtual experience, many HMDs come with speakers or headphones.

Point 6 In 2.2 Personalizing VR:

"hard else or impossible for them" should read "difficult or impossible for them otherwise" difficult or impossible for them otherwise

"student's time a task" has a grammar problem, it should read "student's time to complete a task" or "student's time on task" student's time to complete a task

"Significant effect on the learner's special ability" is unclear.  What special ability? learner’s special needs

"Special education" is inconsistent. Probably should say "Special Education" Special Education

"when designing and production" is inconsistent. Should say "when designing and producing" or possibly "during the design and production"

"a) Determine ... b) Ease of use. c) Analyzing..." is inconsistent. Should rephrase so a) - c) are all verb phrases of the same type, or all noun phrases.

  1. a) Determine the needs of the children.
  2. b)
  3. c) Analyzing the tasks, breaking them down into subtasks, and assigning them out.

"a) Preferred... b) It is preferable" is inconsistent. The a) part should be rephrased as it is not a sentence.

  1. a) The preferred desktop virtual reality system for students with disabilities.
  2. b) A child with learning disabilities would benefit from using a mouse, which is one of the best interaction devices in virtual environments.

"and demonstrate as the following" is unclear and should be rephrased.

According to a study by [12], virtual reality can be used by students with learning disabilities as follows:

"to be violent of written communication" is unclear and should be rephrased.

A child will be less likely to be violent toward written information if attractive colors attract his or her attention.

It occurs twice, in the related work and in your own system requirements.

"be as the bellow" should be rephrased, perhaps just "be:"

be

"a) easy... b) conforms... c) adaptable" is inconsistent, with a) and c) being adjectives but b) being a verb.

  1. a) simple to use and comprehend.
  2. b) follows conventions or standards.
  3. c) flexible and effective in prompting and providing feedback.

In Section 3. Related Work:

"this major Since" should read "this topic since"

this topic since

"This paper discusses A desktop" should read "In this paper, a desktop"

In this paper, a desktop

", [16] This" should read " [16]. This" done

"presents the research examined" should read "presents research that examines"

presents research that examines

"skills,[17]" should read "skills [17]." done

"disabilities [18] This" should read "disabilities [18]. This" done

The discussion of [18] is at present too vague to be useful to this paper.

This paper describes various types of university laboratories from around the world to present the most recent state-of-the-art. A discussion of the educational effectiveness of each laboratory format is also included. Finally, a case study of Technological de Monterrey's development and implementation of virtual reality laboratories and remote labs is presented.

"Needs [19]:" uses all capitalized words for the title of the paper, which is inconsistent with the format of the preceding papers. It uses a colon here, which should either be a period, or all preceding papers should use colon.

In section 4. Methodology:

Capitalization in "Methodology of the proposed Approach" is inconsistent. done

The sentence beginning "The design goal of this system" is awkward and should be rephrased and split into multiple sentences.  A design goal is a different sort of thing than is "establish an interactive communication platform", which sounds more like an overall goal or an implementation goal.

The purpose of this study was to establish an interactive communication platform for teachers and students with learning disabilities in the context of physics experiments, using a 3D virtual physics laboratory.

The sentence beginning "Depending on the scope" is a long run-on sentence that should be broken up into multiple smaller sentences that can be understood.

The sentence beginning "Depending on the scope" is a long run-on sentence that should be broken up into multiple smaller sentences that can be understood.

A system must provide us with the necessary technology and tool to incorporate an emulator of the 3D virtual laboratory (Open Sim) that was used to simulate it. By connecting the virtual environment with the learning management system, Moodle allows students to manage the latter from inside the virtual physics lab.

For the long list of system requirements, it is unclear how much of this is already provided by OpenSim or Sloodle, versus how much you had to invent.

Maybe subdivide requirements into two categories, those that are provided by the underlying off-the-shelf components and those that required development or creative solutions on your part.

4.1. Functional requirements provided by OpenSim or Sloodle:

  1. a) Easily creates and manages educational content of the Learning Management System (Moodle) inside the 3D virtual environment and virtual physics laboratory and interacts with different users in the 3D virtual environment and virtual physics laboratory.
  2. b) The user of the 3D virtual physical laboratory is not allowed to modify the content of the 3D virtual environment, just interact with it.
  3. c) The access authorization and available permissions must allow access to the 3D virtual physical laboratory according to specific authorities, as well as access to the 3D virtual environment itself initially via a username and password.
  4. d) The 3D virtual physical laboratory enables and provides several functions and capabilities to the user and virtual personality including inviting friends to visit his virtual learning physical laboratory and 3D virtual environment, controlling the virtual personality and its appearance, walking, flying, teleporting, sending group messages, listening to group conversations ... Etc.
  5. e) The 3D virtual physical laboratory offers and provides us with a dedicated space for audio conferencing, and sound support and integrates them into the 3D virtual environment, in addition to the automatically integrated sounds compatible with the type of educational 3D virtual environment to enable complete immersion in the experiment, 3D audio effects are necessary.

4.2 Development requirements or creative solutions of our proposed approach:

Our proposed approach manually configures the learning management system to integrate with sloodle tools by copying and pasting sloodle configure files into the learning management system to effectively implement them together, then we create and design the 3D primitive shapes that will be used for experiments, and finally, based on the characteristics of the experiment, we program each object with LSL script. as well as our proposed approach to developing a 3d virtual learning environment contains the following functions:

  1. a) 3D virtual physical laboratory links the name of the user or virtual character in the (Open Sim) emulator to the same user in the learning management system (Moodle) through the email service, as the virtual character enters their account in the learning management system (Moodle) from the 3D virtual environment.
  2. b) Interact with the various tools of Sloodle in the 3D virtual environment.
  3. d) A certain attribute of a 3D virtual physics laboratory is considered a particular criterion for enabling students with disabilities to learn with special content, the criteria included the color for the primitive shapes (balls, cones, cubes, etc.) which were 3D models, the way 3D models fall in accordance with physics laws, students' attention is drawn to interactive sounds in the 3D virtual learning environment, furthermore the flying feature, which enables the user to fly around the 3D virtual laboratory especially when it comes to understanding the rotation and motion of the planet for students with learning disabilities. Below summarized these criteria:
  4. Using attractive colors will catch the child's attention and reduce his or her tendency to be violent of written information.
  5. Using notecards to demonstrate experiments steps and explain it.

iii.       Students with learning disabilities can display presenter tool allow student show whole physics experiment as PDF files, web pages, and videos. 

  1. Students with learning disabilities can use a glossary tool that Moodle stu-dents can use to identify terms. To obtain immediate results, students or teachers must enter the word "/def" followed by the name of the term for which they want to search, which is registered in the glossary.

Parentheses should be removed in "(inviting friends... Etc.)" done

", so to enable" should read ". To enable" done

"the(balls, cones, cubes, etc.)" should read "the primitive shapes (balls, cones, cubes, etc.)"

the primitive shapes (balls, cones, cubes, etc.)"

"the interactive with sound" is unclear and may be missing a word.

Students' attention is drawn to interactive sounds in the 3D virtual learning environment.

"interactive" is an adjective so it requires a following noun.

"attention, furthermore fly feature which enable user fly around" should read "attention. Furthermore, the flying feature, which enables the user to fly around the"

Furthermore, the flying feature, which enables the user to fly around the

"especially when the matter about knowing" is unclear and should be rephrased

especially when it comes to understanding the rotation and motion of the planet for students with learning disabilities

"glossary tool which they can use this tool" is unclear and should be rephrased

a glossary tool that Moodle students can use to identify terms

"architectural style" should be "architecture" architecture

"(Sloodle) software" should be "Sloodle software" Sloodle software

"learning management system (Moodle)" can just be "learning management system" learning management system

In Table 1 there are awkward linebreaks in the 2nd and 5th rows that harm readability. Probably the width of column 1 can be reduced to make column 2 wider and avoid these awkward linebreaks. Alternatively, you could insert linebreaks (after "level" and "capabilities") to avoid the bad ones.

The phrase "learning management software, which is the e-learning method" does not make sense as-written. It should be corrected or clarified.

Lastly, there is the learning management software, which is the delivery method for e-learning.

In Figure 1, "cha" should be "chat" in the right column done

In Section 5. Implementation:

"Figure severalties lab" makes no sense and may be missing words.

As shown in Figure, the 3D virtual lab contained a set of three-dimensional models that could be practiced with, and various experiments were conducted, such as determining the speed at which a fall occurs with of the difference in the reaction of falling. The following three-dimensional models were evaluated for these three-dimensional models, depending on their shape and the height at which they fall: Cube, ball, and cone.

"practice with it a number" makes no sense and may be missing words or punctuation. done

The sentence containing "speed of falling, a number of 3D models ..." makes no sense and should be split into multiple sentences. Within it, the phrase "(cube - A ball - a cone)" should read "such as a cube, a ball, and a cone". That phrase has grammar problems; perhaps a verb is missing after it. done

The sentence containing "The students also liked..." seems overlong and should be split into multiple sentences.

The students also enjoyed the three-dimensional representations, particularly those showing the solar system.

The two sentences "The students also used..." followed by "The students with learning disabilities also used..." seem redundant and should be consolidated. done

"the following stages were implemented in the system according to the following" should read "the system was implemented in the following stages"

the system was implemented in the following stages

"Establishing the virtual lab building in this phase, a lab building" should read "Establishing the virtual lab building. In this phase, a lab building"

Establishing the virtual lab building. In this phase, a lab building

"presentations, we put" should read "presentations. We put"

presentations. We put

The tense used in "would have to become..." and "the two were becoming..." is awkward. Use a more clean past tense to describe how Sloodle is used to integrate tools.

At this stage, the technology configuration files (Sloodle) would have to be a part of the educational management system (Moodle) directories in order for the su-pervisor to add technology activities from within the learning management sys-tem (Moodle),

"the PowerPoint presentation that Established in Moodle chatting tool" makes no sense and should be rephrased.

the Presentation that established in Moodle chatting tool (Web-Intercom) to connect discussion forums in two 3D virtual environments

"fully prepared to use" should read "fully prepared for use" or "fully prepared to be used" fully prepared to be used

In Section 5.2:

"physic laboratory" should read "physics laboratory".  Also, the subsection title should not end with a colon, and should be formatted consistently with Section 5.1, perhaps in italics. done

"experimented with," should read "experiments conducted."

done

"ellipse Mathematically" should read "ellipse. Mathematically"

done

"everybody" should read "every body"

done

In Section 6:

"which they have" should read "that had"

done

"students, Students" should read "students. Students" done

"table 2 below" should read "Table 2 below"; because Table 2 is a proper noun it should be capitalized. done

"fifth hypotheses" should read "five hypotheses"

five hypotheses

The five hypotheses are all missing a required "The" before "first", "second", etc.  Alternately, the phrases "First hypothesis is" through "Fifth hypothesis is" can all be deleted.

done

"(traditional method)" should read "traditional method". This occurs in multiple places. It feels like "(traditional method)" was from some template language that is being copied in here and the authors were supposed to replace it with the method that the control group used.

"e.g table 3" should read "e.g. Table 3". Same correction later for "e.g table 5", "e.g table 6", "e.g table 8", "e.g table 9", "e.g table 11".

done

The names of particular tables are proper nouns and are to be capitalized.

Table 3 contradicts the text that describes it. The text claims that the experimental group performed better when compared to the control group, but the numbers in the table are reversed from the numbers reported in the text. There was mistake and we fixed it.

"In favor of the experimental group which has a higher average." is not a sentence.

In favor of the experimental group which has a higher average.

The phrase "We also calculating" needs to be rephrased in correct English. We are also calculating

Table 5 caption is on a different page from the table; they should be kept together on the same page. done

Table formatting seems to place adequate space between body text and the start of a table, but no space between the end of the table and the following body text. Some lines on tables have unusual marks above or below the line. done

"was ,00000" seems to have an extra comma and maybe a missing period done

Odd break and missing capitalization on line starting "we used the t-test" done

Odd use of parentheses around "(80)%", "(70%)", "(80%)". Remove paretheses. done

"Black, (1.2)" should read "Black (1.2)." Start a following new sentence with "Therefore" done

"Taif. the second" should read "Taif. The second". Similar fix later for "Taif. the third"

Odd break and non-sentence starting "3D virtual physics laboratory in developing" is unclear. Is this the start of a subsection? It needs formatting and/or grammar correction. done

The phrase 'for" Black .The' should read 'for Black. The' done

"the3D" should read "the 3D" (multiple occurrences) done

"sloodle In" should be "sloodle. In".  Should sloodle be capitalized everywhere? done

Similarly, SLOODLE should probably be capitalized Sloodle to be consistent with other uses (multiple occurrences).  Actually sometimes it is referred to as SLoodle.  What spelling did its creators use?  Use that spelling everywhere.

done

"in table 10" should read "in Table 10" done

In Section 7. Discussion of achieved results

Poor linebreaks in Table 11 ("as-" and "ex-". Make that column a little wider.

Table 12 caption is on a separate page from Table 12. Keep them on the same page.

In Section 7.1 The effectiveness of the 3D virtual physics Laboratory, "Laboratory" should not be capitalized when nothing else is. Check journal required format and capitalize section titles as directed.

done

"To assess the effectiveness there" should read "There". The first four words add nothing and should be deleted.

done

Delete the period in "satisfied. with". done

"3d" in item A. should probably be capitalized consistent with other uses: "3D" done

All other occurrences of "3d" should also be fixed. done

Item B contains a long run-on sentence. Split it into multiple sentences.

The second aspect was concerned with interactions. in it described the capabilities of the 3D Virtual Physics Laboratory as very interesting because they were interacting directly with some difficult to interact with objects, such as a solar model.

Sometimes "3D Virtual Physics Laboratory" is all capitals and other times it is "Virtual 3D physics laboratory". The name should be made consistent (both capitalization and whether it is 3D Virtual or Virtual 3D) across all the many uses of this phrase.

done

"table 13 shows" should read "Table 13 shows" done

"figure 4 represents" should read "Figure 4 represents" done

"represents students with learning disabilities answer of many" does not make sense and should be rephrased.

Figure 4 omits much detail and does not show enough information for the reader to draw any conclusions about the student feedback. The supporting body text should elaborate and make clear the language of the various rows for the student choices. done

Reviewer 2 Report

point 1

This is not a response to the points raised. Please correct it.

point 2 

Even though the usefulness of the proposed method is demonstrated by the difference in results from the comparison method, the comparison method is not described in a way that can be reproduced by the reader. The comparison method should be clearly described to clarify the usefulness of the proposed method.

point 3

It is possible that you did not understand my point. In order to show the usefulness of the proposed method in terms of the difference between the results of the proposed and comparative methods, it is necessary to assume that the experimental subjects are the same for the proposed method and the comparative method.

Please provide evidence that the division into an experimental and a control group is done correctly.

point 4

If there is a possibility that this learning system may be effective for people without learning disabilities, that should be described in the paper.

point 7

No corrections have been made. Many corrections remain, such as the use of lowercase letters when they should be capitalized, the use of ( ), where to type periods, and the notation of M_1 in equation (1).

Author Response

Response to Reviewer 2 round2 Comments

Response 1: point 1This is not a response to the points raised. Please correct it. The information provided in 1. Introduction gives the impression that it is the opinion of the authors. Please cite the literature properly and explain logically based on evidence.

Introduction

Virtual learning environments (VLEs), is learning tools that incorporate computers and the Internet into instruction to enhance a student's learning experience Further-more they are increasingly being used in a variety of settings, including classrooms, in-formal learning environments, distance learning, business, and many others. In a vir-tual world based on a computer-simulated 3D environment, users interact with ava-tars. This habitat is typically represented in two- or three-dimensional forms in graph-ical representations of humanoids [1]. Virtual environments, in addition to being in-teractive, engaging, and safe, typically provide students with immediate feedback, al-lowing them to learn by doing. Because of their importance and potential, 3D virtual platforms created with the OpenSimulation tool rank high among educational virtual environments. The ability to integrate knowledge and skills with educational VLEs and exchange information with intellectual information management will be critical in in-creasing the adoption of virtual reality. On the other hand VR can assist students with disabilities in developing their knowledge, abilities, and behaviors in different manners which would not have been possible otherwise, allowing them to engage in academic tasks that are largely free of the restrictions set by their disability and totally secured. VR also tends to help everyone else feel compassion for people with disabilities by ena-bling them to practice disabilities through virtual worlds [2]. According to American psychiatric association the classification of the students with disabilities are ten cate-gories: intellectual disabilities, autism spectrum disorder (ASD), attention-deficit/hyperactivity disorder, global developmental delay, communication disorders, specific learning disorders, developmental coordination disorders, stereotypical movement disorders, tic disorders, and Tourette’s syndrome. Among these disorders, the highest rates are seen for intellectual disability and ASD [3] [26][28]. As well as of-fering a compelling and exciting experience, these applications help students with learning disabilities reduce the impact of their disabilities, enhance their quality of life, get involved in societies, and learn how to live, be more accessible, and develop their cognitive abilities [4]. The design of virtual environments (VEs) can be adapted to meet the needs of students with a wide range of educational backgrounds, physical capabili-ties, verbal capabilities, and cognitive abilities [2]. In contrast, students with learning disabilities can discover or build new worlds or control items without being restricted by their disabilities if they are provided with an appropriate interface, by giving stu-dents a sense of control in their surroundings through virtual reality's freedom of mo-bility [4]. A VE is more personal than most real-life environments because it can be tai-lored to the abilities of the individual, allowing individuals with disabilities to partici-pate [4]. It is possible to use virtual reality to engage students with learning disabilities, focus on their talents, and enable them to gain a sense of mastery [5]. In our paper, we are therefore attempting to prove that virtual reality can benefit students with learn-ing disabilities.

point 2

Even though the usefulness of the proposed method is demonstrated by the difference in results from the comparison method, the comparison method is not described in a way that can be reproduced by the reader. The comparison method should be clearly described to clarify the usefulness of the proposed method.

We created our virtual physics laboratory based on the simulation model repository files (Open Simulator) installed through our simulation software file system. Pre- and post-tests were conducted on the same subjects and physical experiments for both groups of students with learning disabilities. As a result of using the 3D virtual physics laboratory, female students with learning disabilities performed well in the study, showing how effective virtual reality laboratories are for learning disabled students, and illustrates the effectiveness of the SLOODLE 3D Virtual Physics Laboratory as shown in Example Table 11 and Example Table 12.:

Subject

Aspect

Results

Experimental group's

Control group's

Post-application

cognitive test

15.50

11.15

Performance

101.30

74.40

Practical skills

116.80

22.42

Proficiency and mastery

Degree of mastery 80%

98.0%

Degree of proficiency 80%

70.0%

point 3

It is possible that you did not understand my point. In order to show the usefulness of the proposed method in terms of the difference between the results of the proposed and comparative methods, it is necessary to assume that the experimental subjects are the same for the proposed method and the comparative method.

Please provide evidence that the division into an experimental and a control group is done correctly.. Students with learning disabilities were divided into two groups equal groups (n = 20 each) based on their disability type.

point 4

If there is a possibility that this learning system may be effective for people without learning disabilities, that should be described in the paper.

 It may be appropriate for individuals who do not have learning disabilities to use the 3D virtual physics lab, since they can learn while having fun and test physics experiments from a different point of view, for instance with the solar system model experiment, students can get to know the planets up close, As a result, students without learning disabilities can develop their cognitive and skills.

point 7

No corrections have been made. Many corrections remain, such as the use of lowercase letters when they should be capitalized, the use of ( ), where to type periods, and the notation of M_1 in equation (1). done

Round 3

Reviewer 2 Report

The content of the paper has been improved, but typographical errors remain. Please revise your paper carefully.

M2 in equation (5) has subscripts. Please check if this is not a problem.

There are many typographical errors.

e.g.

b) It interacts with the various tools of SLOODLE in the 3D virtual environment. d) A certain attribute of a 3D virtual physics laboratory considers a particular criterion...

Virtual Learning Environments (VLEs) is learning...

According tothe study ...

Author Response

Response to Reviewer 2 round3 Comments

Thank you very much for your appreciates remarks

Response 1: M2 in equation (5) has subscripts. Please check if this is not a problem

Point 2 : There are many typographical errors.

  1. b) It interacts with the various tools of SLOODLE in the 3D virtual environment. d) A certain attribute of a 3D virtual physics laboratory considers a particular criterion...

Virtual Learning Environments (VLEs) is learning...According tothe study ...

  1. The 3D virtual physical laboratory links the name of the user or virtual character in the (Open Sim) emulator to the same user in the learning management system (Moodle) through the email service, as the virtual character enters their account in the learning management system (Moodle) from the 3D virtual environment.
  2. It interacts with the various tools of SLOODLE in the 3D virtual environment.
  3. A certain attribute of a 3D virtual physics laboratory considers a particular criterion for enabling students with disabilities to learn with special content. The criteria include the color for the primitive 3D-model shapes (balls, cones, cubes, etc.), the way 3D models fall in accordance with physics laws, students' attention to interactive sounds in the 3D virtual learning environment, and the flying features which enable the user to fly around the 3D virtual laboratory especially when it comes to understanding the rotation and motion of the planet for students with learning disabilities.

 The 3D virtual environment allows disabled student avatars to interact with SLOODLE tools.

 3D virtual physics laboratory have a particular attribute designed to facilitate learning for students with disabilities. A number of criteria are considered, such as the color of primitive 3D-model shapes (balls, cones, cubes, etc.), the way the 3D models fall in accordance with physics laws, students' awareness of interactive sounds in the 3D virtual learning environment, and the ability to fly around in the 3D virtual laboratory, especially for students with learning disabilities who need to understand the rotation and motion of the planet.
